

# MSWEP: 3-hourly 0.25° global gridded precipitation (1979–2015) by merging gauge, satellite, and reanalysis data

Hylke E. Beck[1], Albert I.J.M. van Dijk[2], Vincenzo Levizzani[3], Jaap Schellekens[4], Diego G. Miralles[5,6], Brecht Martens[5], and Ad de Roo[1]

[1]European Commission, Joint Research Centre (JRC), Via Enrico Fermi 2749, 21027 Ispra (VA), Italy
[2]Fenner School of Environment & Society, The Australian National University, Canberra, Australia
[3]National Research Council of Italy, Institute of Atmospheric Sciences and Climate (CNR-ISAC), Bologna, Italy
[4]Inland Water Systems Unit, Deltares, Delft, The Netherlands
[5]Laboratory of Hydrology and Water Management, Ghent University, Ghent, Belgium
[6]Department of Earth Sciences, VU University Amsterdam, Amsterdam, the Netherlands

*Correspondence to:* Hylke E. Beck (hylke.beck@gmail.com)

**Abstract.** Current global precipitation ($P$) datasets do not take full advantage of the complementary nature of satellite and reanalysis data. Here, we present Multi-Source Weighted-Ensemble Precipitation (MSWEP), a global $P$ dataset for the period 1979–2015 with a 3-hourly temporal and 0.25° spatial resolution, specifically designed for hydrological modeling. The design philosophy of MSWEP was to optimally merge the highest quality $P$ data sources available as a function of time scale and

location. The long-term mean of MSWEP was based on the CHPclim dataset but replaced with more accurate regional datasets where available. A correction for gauge under-catch and orographic effects was introduced by inferring catchment-average $P$ from streamflow ($Q$) observations at 13 762 stations across the globe. The temporal variability of MSWEP was determined by weighted averaging of $P$ anomalies from seven datasets; two based solely on interpolation of gauge observations (CPC Unified and GPCC), three on satellite remote sensing (CMORPH, GSMaP-MVK, and TMPA 3B42RT), and two on atmospheric model

reanalysis (ERA-Interim and JRA-55). For each grid cell, the weight assigned to the gauge-based estimates was calculated from the gauge network density, while the weights assigned to the satellite- and reanalysis-based estimates were calculated from their comparative performance at the surrounding gauges. The quality of MSWEP was compared against four state-of-the-art gauge-adjusted $P$ datasets (WFDEI-CRU, GPCP-1DD, TMPA 3B42, and CPC Unified) using independent $P$ data from 125 FLUXNET tower stations around the globe. MSWEP obtained the highest daily correlation coefficient ($R$) among the five

$P$ datasets for 60.0 % of the stations and a median $R$ of 0.67 versus 0.44–0.59 for the other datasets. We further evaluated the performance of MSWEP using hydrological modeling for 9011 catchments ($< 50\,000$ km$^2$) across the globe. Specifically, we calibrated the simple conceptual hydrological model HBV against daily $Q$ observations with $P$ from each of the different datasets. For the 1058 sparsely-gauged catchments, representative of 83.9 % of the global land surface (excluding Antarctica), MSWEP obtained a median calibration NSE of 0.52 versus 0.29–0.39 for the other $P$ datasets. MSWEP is available via

http://www.gloh2o.org.



## 1 Introduction

A quantitative appraisal of precipitation ($P$) amount and of its spatio-temporal distribution is essential for many scientific and operational applications, including, but not limited to: increasing our understanding of the hydrological cycle; assessing the hydrological impacts of human activities; assessing water resources; irrigation planning; and forecasting of droughts and floods (Golding, 2009; Kucera et al., 2013; Pozzi et al., 2013; Serrat-Capdevila et al., 2013; Lettenmaier et al., 2015). However, $P$ is also one of the meteorological variables that is most difficult to estimate, due to its high spatio-temporal heterogeneity (Daly et al., 1994; Herold et al., 2015). There are currently four principal measurement or modeling sources to determine $P$ (Michaelides et al., 2009): (i) ground-based gauge observations; (ii) ground-based radar remote sensing, (iii) satellite remote sensing; and (iv) atmospheric retrospective-analysis models.

Due to the highly localized nature of gauge observations, the accuracy of interpolation-based spatial $P$ estimates is highly dependent on gauge network density and the degree of spatial coherence, both of which are highly variable globally (Krajewski et al., 2003; Hijmans et al., 2005; Chen et al., 2008). In addition, gauge observations are typically affected by systematic biases in mountainous environments due to the elevation bias in gauge placement and in snow-dominated regions due to wind-induced under-catch (Groisman and Legates, 1994; Rasmussen et al., 2012). Ground based radars offer high temporal and spatial resolution but their coverage is limited to developed regions (Koistinen, 1991; Kitchen and Blackall, 1992; Kitchen et al., 1994; Krajewski and Smith, 2002; Martens et al., 2013). Moreover, very few merged radar data products are readily available.

Satellites are capable of observing large areas instantaneously at a high resolution. They are particularly suitable for rainfall estimation in the tropics, which exhibit highly heterogeneous rainfall patterns due to the importance of convective storms (Smith et al., 2005). However, satellite retrieval approaches are susceptible to systematic biases, relatively insensitive to light rainfall events, and tend to fail over snow- and ice-covered surfaces (Ferraro et al., 1998; Ebert et al., 2007; Kidd and Levizzani, 2011; Kidd et al., 2012; Laviola et al., 2013). Finally, atmospheric reanalysis models are ideally suited for simulating the evolution of large-scale (synoptic) weather systems, but poorly represent the variability associated with convection due mainly to their relatively low resolution and deficiencies in the parameterizations of sub-grid processes (Roads, 2003; Ebert et al., 2007; Kidd et al., 2013). Only recently cloud-resolving models have started to tackle the convection-$P$ link (e.g., Petch, 2004; Li and Gao, 2012). Although only available regionally, ground-based gauges and radars are generally regarded as the most reliable way of estimating $P$ and thus have frequently been used for evaluation and improvement of satellite- and reanalysis-based $P$ estimates (Maggioni et al., 2016).

Table 1 lists a selection of 21 publicly available gridded $P$ datasets suitable for (quasi-)global hydrological studies (for more exhaustive overviews, see http://ipwg.isac.cnr.it and http://reanalyses.org). The datasets vary in terms of spatial resolution (from $0.1°$ to $2.5°$), spatial coverage (from $< 50°$N/S to global), temporal resolution (from 30 minutes to monthly), temporal span (from 1 to 114 years), and data sources employed (gauge, satellite, and reanalysis, and combinations thereof). TMPA 3B42RT, for example, combines $P$ estimates derived from passive microwave (PMW) and thermal infrared (IR) observations from multiple satellite sensors with radar data from the Tropical Rainfall Measuring Mission (TRMM; Huffman et al., 2007), while





CMORPH propagates PMW-based $P$ estimates using motion vectors derived from IR data (Joyce et al., 2004). Conversely, CHIRPS (Funk et al., 2015a) and TMPA 3B43 (Huffman et al., 2007) combine satellite and gauge $P$ estimates using inverse-error weighted averaging. PFD (Sheffield et al., 2006) and WFDEI (Weedon et al., 2014) rescale reanalysis $P$ estimates to force agreement with gauge-interpolated $P$ estimates on a monthly basis (although such rescaling is unlikely to yield improvements in regions with sparse gauge networks). Other merging techniques include multivariate analysis (e.g., Funk et al., 2015b), probability distribution analysis (e.g., Anagnostou et al., 1999), geostatistical estimators (e.g., Seo, 1998; Grimes et al., 1999), wavelet analysis (e.g., Heidinger et al., 2012), and Bayesian methods (e.g., Todini, 2001).

The datasets have been designed for different applications and provide sometimes widely varying $P$ estimates (Adler et al., 2001; Bosilovich et al., 2008; Kucera et al., 2013; Skok et al., 2015; Prein and Gobiet, 2016), even among gauge-adjusted datasets (Herold et al., 2015). Despite several studies that intercompared and evaluated these $P$ datasets in different regions (for non-exhaustive overviews, see Serrat-Capdevila et al., 2013 and Maggioni et al., 2016), so far no clear consensus has emerged on which estimation approach is superior overall. Indeed, each approach comes with potential disadvantages that may restrict its usefulness for global hydrological modeling. Common issues include:

1. With the exception of CMAP, all datasets listed in Table 1 employ only one or two of the main data sources—either gauge and satellite, or gauge and reanalysis—and thus no not take full advantage of the complementary nature of satellite and reanalysis data identified in previous studies (e.g., Janowiak, 1992; Huffman et al., 1995; Xie and Arkin, 1996, 1997; Adler et al., 2001; Ebert et al., 2007; Serrat-Capdevila et al., 2013; Peña Arancibia et al., 2013; Xie and Joyce, 2014).

2. Many of the listed datasets do not explicitly and fully account for gauge under-catch and/or orographic effects, and consequently underestimate $P$ in many regions around the globe (e.g., Zaitchik et al., 2010; Kauffeldt et al., 2013; Beck et al., 2015, 2016a; Prein and Gobiet, 2016).

3. Many datasets (numbered 6–21 in Table 1) do not incorporate gauge observations or do so on a monthly basis, and hence may not make optimal use of valuable information on the daily $P$ variability provided by gauges.

4. Some datasets (1–5, 8, 13, 16, 18, 20, and 21) have a spatial resolution that is much coarser ($\geq 0.5°$) than desirable for hydrological applications, given the importance of non-linear responses to rainfall.

5. Several datasets (6, 7, 9–12, and 19–20) have a quasi-global spatial coverage, precluding truly global terrestrial applications.

6. A few of the datasets (2–5 and 21) have a monthly temporal resolution which is insufficient for most dynamic hydrological modeling applications.

7. Finally, several datasets (7, 9, 10, 12, 19, and 20) cover a relatively short period ($< 20$ years), which is problematic when assessing hydrological changes in a long-term context.

This research was motivated by the opportunity to develop a global gridded $P$ dataset with a 3-hourly temporal and $0.25°$ spatial resolution for the period 1979–2015, characteristics that make it more suitable for hydrological applications for the





above-mentioned reasons. The dataset has been named Multi-Source Weighted-Ensemble Precipitation (MSWEP) as it seeks optimally combine data from various gauge, satellite, and reanalysis data sources.

## 2 MSWEP methodology

Figure 1 presents a flow chart summarizing the main steps carried out to produce MSWEP. The methodology described here
corresponds to version 1.0 of the dataset. In brief, we first derived a long-term bias-corrected climatic mean (explained in Section 2.1). Next, several gridded satellite and reanalysis $P$ datasets were evaluated in terms of temporal variability to asses their potential inclusion in MSWEP (explained in Section 2.2). The long-term climatic mean was subsequently temporally downscaled in a stepwise manner first to the monthly, then to the daily, and finally to the 3-hourly time scale using weighted averages of $P$ anomalies derived from gauge, satellite, and reanalysis datasets to yield the final MSWEP dataset (explained in
Section 2.3).

### 2.1 Bias correction of CHPclim

The long-term mean of MSWEP was based on the recently released Climate Hazards Group Precipitation Climatology (CHPclim) dataset ($0.05°$ resolution; version 1.0; Funk et al., 2015b), a global $P$ climatology based on gauge observations and satellite data. The CHPclim data were replaced with more accurate Parameter-elevation Relationships on Independent Slopes
Model (PRISM) data for the conterminous USA (1-km resolution; Daly et al., 1994) and Tait et al. (2006) data for New Zealand ($0.05°$ resolution). For brevity we refer to this combination of gauge-analysis products as CHPclim however. Although CHPclim has been corrected for orographic effects it has not been adjusted for wind-induced gauge under-catch, and is thus expected to underestimate the actual $P$, especially in snow-dominated regions. Two approaches were tested to correct for this.

### 2.1.1 Bias correction using catch-ratio equations

The first approach uses country-specific catch-ratio (CR) equations for snowfall from Goodison et al. (1998), as summarized by Adam and Lettenmaier (2003), and the rainfall CR equation for unshielded gauges from Yang et al. (1998). The mean wind speed data required as input were derived from a global 5-km map of mean wind speed at 80-m height based on atmospheric reanalysis model output (Vaisala, 2015). To estimate the wind speed value at 5-m height we used the wind profile power-law relationship defined as:

$$v = v_r \left( z/z_r \right)^a, \tag{1}$$

where $v$ (m/s) represents the wind speed at height $z$ (m), and $v_r$ (m/s) represents the wind speed at reference height $z_r$ (m). The exponent $a$ (unitless) is an empirical coefficient that depends on surface roughness and atmospheric stability and was set to 0.3 following Irwin (1979). A global map of the snowfall fraction of $P$ was computed based on air temperature ($T_a$) data from the gauge-based WorldClim monthly climatic dataset (1-km resolution; version 1.4 release 3; Hijmans et al., 2005) using
a $T_a$ threshold of $1°C$ to distinguish between rain and snow. It is noted that the CR equations were not designed for application

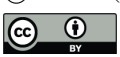



**Table 1.** Non-exhaustive overview of freely available (quasi-)global gridded $P$ datasets. If a particular dataset is available in different spatial resolutions or in different variants we only listed the 'best' one (e.g., we list the gauge-adjusted 3B42 variant of TMPA rather than the non-gauge adjusted 3B42RT variant). The datasets are sorted first by source and then alphabetically by short name. MSWEP has been added for the sake of completeness.

| | Short name | Full name and details | Data source(s) | Spatial resolution | Spatial coverage | Temporal resolution | Temporal coverage | Reference(s) |
|---|---|---|---|---|---|---|---|---|
| 1 | CPC Unified | Climate Prediction Center (CPC) Unified | Gauge | $0.5°^{a}$ | Global | Daily | 1979–present | Xie et al. (2007); Chen et al. (2008) |
| 2 | CRU | Climatic Research Unit (CRU) Time-Series (TS) | Gauge | $0.5°$ | Global | Monthly | 1901–2014 | Harris et al. (2013) |
| 3 | GPCC | Global Precipitation Climatology Centre (GPCC) Full Data Reanalysis and First Guess | Gauge | $0.5°^{b}$ | Global | Monthly | 1901–present | Schneider et al. (2014) |
| 4 | PREC/L | PRECipitation REConstruction over Land (PREC/L) | Gauge | $0.5°$ | Global | Monthly | 1948–present | Chen et al. (2002) |
| 5 | UDEL | University of Delaware (UDEL) | Gauge | $0.5°$ | Global | Monthly | 1901–2014 | Matsuura and Willmott (2009) |
| 6 | CHIRPS | Climate Hazards group Infrared Precipitation with Stations (CHIRPS) | Gauge, satellite | $0.05°$ | 50°N–50°S | Daily | 1981–present | Funk et al. (2015a) |
| 7 | CMORPH | CPC MORPHing technique (CMORPH) | Gauge, satellite | $0.25°$ | 60°N–60°S | Daily | 1998–present | Joyce et al. (2004) |
| 8 | GPCP-1DD | Global Precipitation Climatology Project (GPCP) 1-Degree Daily (1DD) Combination | Gauge, satellite | $1°$ | Global | Daily | 1996–2015 | Huffman et al. (2001) |
| 9 | GSMaP-MVK | Global Satellite Mapping of Precipitation (GSMaP) Moving Vector with Kalman (MVK) | Gauge, satellite | $0.1°$ | 60°N–60°S | Hourly | 2000–present | Iguchi et al. (2009) |
| 10 | IMERG | Integrated Multi-satellitE Retrievals for GPM (IMERG) | Gauge, satellite | $0.05°$ | 70°N–70°S | 30 minutes | 2014–present | Huffman et al. (2014) |
| 11 | PERSIANN-CDR | Precipitation Estimation from Remotely Sensed Information using Artificial Neural Networks (PERSIANN) Climate Data Record (CDR) | Gauge, satellite | $0.25°$ | 60°N–60°S | 6 hourly | 1983 to 2012 | Ashouri et al. (2015) |
| 12 | TMPA 3B42 | TRMM Multi-satellite Precipitation Analysis (TMPA) 3B42 | Gauge, satellite | $0.25°$ | 50°N–50°S | 3 hourly | 1998–present | Huffman et al. (2007) |
| 13 | MERRA-Land | Modern Era Retrospective-Analysis for Research and Applications (MERRA)-Land | Gauge, reanalysis | $0.5° × 0.67°$ | Global | Hourly | 1979–present | Reichle et al. (2011) |
| 14 | PFD | Princeton global meteorological Forcing Dataset | Gauge, reanalysis | $0.25°$ | Global | 3 hourly | 1948–2012 | Sheffield et al. (2006) |
| 15 | WFDEI | WATCH Forcing Data ERA-Interim (WFDEI) | Gauge, reanalysis | $0.25°$ | Global | 3 hourly | 1979–2014 | Weedon et al. (2014) |
| 16 | NCEP-CFSR | National Centers for Environmental Prediction (NCEP) Climate Forecast System Reanalysis (CFSR) | Reanalysis | $0.3125°^{c}$ | Global | Hourly | 1979 to 2010 | Saha et al. (2010) |
| 17 | ERA-Interim | European Centre for Medium-range Weather Forecasts ReAnalysis Interim (ERA-Interim) | Reanalysis | $0.25°^{c}$ | Global | 3 hourly | 1979–2014 | Dee et al. (2011) |
| 18 | JRA-55 | Japanese 55-year ReAnalysis (JRA-55) | Reanalysis | $1.25°$ | Global | 3 hourly | 1959–present | Kobayashi et al. (2015) |
| 19 | CHOMPS | Cooperative Institute for Climate Studies (CICS) High-Resolution Optimally Interpolated Microwave Precipitation from Satellites (CHOMPS) | Satellite | $0.25°$ | 60°N–60°S | Daily | 1998–2007 | Joseph et al. (2009) |
| 20 | SM2RAIN-ASCAT | Based on Advanced Scatterometer (ASCAT) data (Brocca et al., 2016) | Satellite | $0.5°$ | Global | Daily | 2007–2015 | Brocca et al. (2014) |
| 21 | CMAP | CPC Merged Analysis of Precipitation (CMAP) | Gauge, satellite, re-analysis | $2.5°$ | Global | 5 day | 1979–present | Xie and Arkin (1996, 1997) |
| 22 | MSWEP | Multi-Source Weighted-Ensemble Precipitation (MSWEP) | Gauge, satellite, re-analysis | $0.25°$ | Global | 3 hourly | 1979–2015 | This study |

[a] 0.25° spatial resolution for the conterminous USA.

[b] 1° spatial resolution for 2014–present.

[c] ~80 km effective spatial resolution (i.e., the resolution of the employed atmospheric model).



**Figure 1.** Flow chart summarizing the main steps carried out to produce MSWEP. The different colors represent different time scales. $P_n$ refers to normalized $P$ (i.e., $P$ anomalies).





at monthly time scales. However, the necessary daily station observations of $P$, $T_a$, and wind speed are unfortunately generally not available in mountainous regions where they are needed the most (Viviroli et al., 2007).

### 2.1.2 Bias correction based on $Q$ observations

The second approach tested to bias correct CHPclim involves the use of observation-based estimates of long-term streamflow
($Q$) and potential evaporation ($E_p$) to infer the 'true' $P$ (cf. Adam et al., 2006; Henn et al., 2015). For this purpose we employed the Zhang et al. (2001) relationship, which has been frequently used to estimate long-term actual evaporation ($E$) from long-term $P$ and $E_p$ values. Here we use it in an inverse manner to estimate, only for regions with snowfall and/or complex topography, the underestimation of long-term $P$ from long-term $Q$ and $E_p$ data under the assumption that the long-term $E$ equals the difference between long-term $P$ and $Q$ and that changes in water storage are negligible. The Zhang et al.
(2001) relationship was chosen over several other similar relationships (Ol'dekop, 1911; Pike, 1964; Budyko, 1974; Porporato et al., 2004) because it includes an empirical parameter related to the plant-available water capacity termed $w$ (unitless). This parameter was set to the "shortgrass and crops" value of 0.5 to produce a conservative correction of $P$. The Zhang et al. (2001) relationship is given by:

$$\frac{E}{P} = \frac{1 + w\frac{E_p}{P}}{1 + w\frac{E_p}{P} + \left(\frac{E_p}{P}\right)^{-1}},$$ (2)

where $P$, $E_p$, and $Q$ are in units mm yr$^{-1}$. Assuming that $E = P - Q$, the Zhang et al. (2001) relationship reformulated to yield $P$ is given by:

$$P = \frac{Q}{3} + \sqrt[3]{-\frac{b}{2} + \sqrt{\frac{b^2}{4} + \frac{a^3}{27}}} + \sqrt[3]{-\frac{b}{2} - \sqrt{\frac{b^2}{4} + \frac{a^3}{27}}},$$ (3)

where $a$ and $b$ are defined as:

$$a = -\frac{Q^2}{3} - QE_p \qquad \text{and}$$ (4)

$$b = \frac{2Q^3}{27} - \frac{Q^2 E_p}{3} - wE_p^2 Q,$$ (5)

respectively.

Following this method, we inferred long-term $P$ from $Q$ records for 13 762 regions (12 233 catchments and 1529 interstation catchment areas) across the globe. Long-term $Q$ estimates for these catchments were computed from the same three sources as those used by Beck et al. (2015), namely (i) the US Geological Survey (USGS) Geospatial Attributes of Gages for Evaluating
Streamflow (GAGES)-II database (Falcone et al., 2010), (ii) the Global Runoff Data Centre (GRDC; http://www.bafg.de/ GRDC/), and (iii) the Australian Peel et al. (2000) database. We only used catchments $< 10\,000$ km$^2$ with a $Q$ record length $> 3$ year (not necessarily consecutive). The long-term $Q$ estimates for the interstation catchment areas were obtained from a map of long-term $Q$ based on observations from 1651 large catchments (10 003–4 691 000 km$^2$) worldwide (Beck, 2016; version 1.2).





To assess the sensitivity of the $P$ values inferred using Equation 3 to biases in $E_\mathrm{p}$, we used $E_\mathrm{p}$ estimates derived using three formulations: (i) the temperature-based Hargreaves and Samani (1982) equation, (ii) the radiation-based Priestley and Taylor (1972) approach, and (iii) the FAO-56 Penman-Monteith combination equation (Allen et al., 1998). The $E_\mathrm{p}$ data were produced by Sperna Weiland et al. (2015) using meteorological inputs from the WFDEI dataset. Since the resulting $P$ values generally differed by $< 10\,\%$ depending on the $E_\mathrm{p}$ formulation the mean of the three estimates was adopted.

Next, we computed bias correction values for each catchment area as the ratio of $P$ inferred from Equation 3 over the original, unadjusted $P$. The resulting values were subsequently interpolated to produce a continuous global map by calculating, for each $0.25°$ grid cell with snowfall and/or complex topography, the median bias correction factor of the 10 'closest' catchments. Note that in calculating the median we also included catchments with bias correction factors $< 1$. An empirical distance measure was developed as:

$$d = \sqrt{x^2 + y^2 + z^2}, \tag{6}$$

where $y$ (km) is the longitudinal distance to the catchment centroid, $x$ (km) the latitudinal distance to the catchment centroid, and $z$ (m) the difference between the grid-cell mean elevation and the catchment-mean elevation (note the different horizontal and vertical length units). By including $z$, the bias correction factors obtained for mountainous catchments exert less influence on the adjacent flat terrain. By using the median the impact of potential outlier errors in the $Q$ observations, $E_\mathrm{p}$ estimates, and catchment boundary data is limited. Bias correction factors were only calculated for regions with snowfall, defined by a snowfall fraction $> 0.25$, and/or complex topography, defined by an "effective terrain height" $< 100$ m (Daly et al., 2008). The snowfall fraction was computed using WorldClim $T_\mathrm{a}$ data using a threshold of $1°C$, while the elevation data necessary for calculating $z$ and the effective terrain height were derived from Consultative Group for International Agricultural Research (CGIAR) Shuttle Radar Topography Mission (SRTM) data (90-m resolution; version 4.1; Farr et al., 2007) for latitudes $< 60°N$ and GTOPO30 ($1°$ resolution; http://lta.cr.usgs.gov/GTOPO30) for latitudes $> 60°N$.

## 2.2 Gauge-based evaluation of gridded $P$ datasets

Eight non-gauge adjusted satellite and reanalysis $P$ datasets were evaluated in terms of temporal variability to asses their potential inclusion in MSWEP (see Table 2). These datasets were specifically chosen for their lack of gauge adjustments, allowing us to evaluate their performance using gauge observations. For the evaluation, we considered daily $P$ data from 99 444 gauges included in the Global Historical Climatology Network-Daily (GHCN-D) database (Gleason, 2002) and 16 912 gauges included in the Global Summary of the Day (GSOD) database (amounting to 116 356 stations in total). All gridded $P$ data were resampled to $0.5°$ for consistency, and if there were multiple gauges located within a single $0.5°$ grid cell the mean gauged $P$ was calculated. Only stations with $> 4$ months of data (not necessarily consecutive) between March 2000 (marking the start of GSMaP-MVK and PERSIANN) and December 2010 (marking the end of GSMaP-MVK version 5) were used, and if a particular station was included in both databases we only kept the one in the GHCN-D.

Previous $P$ evaluation studies have used a large variety of evaluation metrics (see the overview by Gebremichael, 2010). Here we use Pearson correlation coefficients computed between 3-day mean rainfall and snowfall time series from the gauges





**Table 2.** Overview of the (quasi-)global $P$ datasets considered for inclusion in MSWEP.

| Short name | Full name and details | Data source | Spatial resolution | Temporal resolution | Spatial coverage | Temporal span | Reference |
|---|---|---|---|---|---|---|---|
| CMORPH* | CPC MORPHing technique (CMORPH) version 1.0 | Satellite | 0.25° | 3 hourly | 60°N–60°S | Jan-1998 to Jan-2015 | Joyce et al. (2004) |
| GSMaP-MVK* | Global Satellite Mapping of Precipitation (GSMaP) Moving Vector with Kalman (MVK) standard versions 5 and 6 | Satellite | 0.1° | Hourly | 60°N–60°S | Mar-2000 to Dec-2010; Mar-2014 to Oct-2015 | Iguchi et al. (2009) |
| TMPA 3B42RT* | TRMM Multi-satellite Precipitation Analysis (TMPA) 3B42RT version 7 | Satellite | 0.25° | 3 hourly | 50°N–50°S | Jan-2000 to Oct-2015 | Huffman et al. (2007) |
| PERSIANN | Precipitation Estimation from Remotely Sensed Information using Artificial Neural Networks (PERSIANN) | Satellite | 0.25° | 3 hourly | 60°N–60°S | Mar-2000 to Feb-2015 | Sorooshian et al. (2000) |
| SM2RAIN-ASCAT | Based on ASCAT data (Brocca et al., 2016) | Satellite | 0.5° | Daily | Global | 2007–2015 | Brocca et al. (2014) |
| NCEP-CFSR | National Centers for Environmental Prediction (NCEP) Climate Forecast System Reanalysis (CFSR) | Reanalysis | 0.31° | Hourly | Global | Jan-1979 to Mar-2011 | Saha et al. (2010) |
| ERA-Interim* | European Centre for Medium-range Weather Forecasts ReAnalysis Interim (ERA-Interim) | Reanalysis | 0.25° [a] | 3 hourly | Global | Jan-1979 to Dec-2014 | Dee et al. (2011) |
| JRA-55* | Japanese 55-year ReAnalysis (JRA-55) | Reanalysis | 1.25° | 3 hourly | Global | Jan-1959 to present | Kobayashi et al. (2015) |
| CPC Unified* | Climate Prediction Center (CPC) Unified version 1.0 and real time | Gauge | 0.5° [b] | Global | Daily | 1979–present | Xie et al. (2007); Chen et al. (2008) |
| GPCC* | Global Precipitation Climatology Centre (GPCC) Full Data Reanalysis and First Guess version 7 | Gauge | 0.5° [c] | Global | Monthly | 1901–present | Schneider et al. (2014) |

[*] Selected to be included in MSWEP.

[a] ~80 km effective spatial resolution (i.e., the resolution of the employed atmospheric model).

[b] 0.25° spatial resolution for the conterminous USA.

[c] 1° spatial resolution for 2014–present.

and the gridded datasets. Correlation coefficients are highly sensitive to large $P$ events, which is a desirable feature since large $P$ events are more important in the context of hydrological modeling. We used 3-day averages instead of daily averages to account for mismatches in the 24-hour measurement period between the datasets and the gauges and to reduce the influence of erroneous measurements. Snowfall and rainfall were evaluated separately, and were distinguished using a $T_a$ threshold of

5    $1°$C based on daily mean $T_a$ data from the ERA-Interim atmospheric reanalysis dataset (Dee et al., 2011) resampled to 0.5° using nearest neighbor and offset to match the long-term mean of WorldClim. Gauges with correlation coefficients $< 0.4$ for all datasets were deemed unreliable and excluded from the analysis (cf. Collischonn et al., 2008). The performance of each $P$ dataset in terms of bias was not evaluated since MSWEP relies on CHPclim for its long-term average.

## 2.3    Merging procedure

10    The procedure to generate the MSWEP dataset involves four distinct stages of merging (see Figure 1). In the first, we produced a 0.25° global map of long-term mean $P$ based on the CHPclim dataset replaced with more accurate regional datasets for the USA and New Zealand and corrected for bias (see Section 2.1).





In the second merging stage (see Figure 1), for the monthly, daily, and 3-hourly time scales separately, and for the satellite and reanalysis datasets separately, we merged $P$ anomalies from the different datasets by weighted averaging to yield one $P$ anomaly estimate at each time scale for each data source. The weight maps for each dataset were derived for rainfall and snowfall separately from the 3-day correlation coefficients obtained in the evaluation using gauge observations (see Section 2.2).

The resulting 3-day correlation coefficients were first squared to yield the explained variance and subsequently interpolated to produce continuous global weight maps by calculating, for each $0.25°$ grid cell, the median of the 10 most nearby gauges. The rainfall interpolated weight was used for grid cells with $T_a > 3°C$, while the snowfall interpolated weight was used for grid cells with $T_a < 3°C$. The weights for satellite data were set to 0 for grid cells with $T_a < 3°C$. $T_a$ data were taken from ERA-Interim resampled to $0.25°$ using nearest neighbor and offset to match the long-term means of WorldClim.

In the third merging stage (see Figure 1), for the monthly, daily, and 3-hourly time scales separately, we merged the $P$ anomalies from the different data sources by weighted averaging to yield one $P$ anomaly estimate at each time scale. For the satellite and reanalysis data sources, the weights were set to the maximum of the weights assigned to the individual datasets in the second merging stage. For example, if for a particular grid cell the GSMaP-MVK dataset was assigned the highest weight in the second merging stage, we used the GSMaP-MVK weight for the satellite data source in the third merging stage. Weights

for the gauge data were empirically calculated as a function of gauge density by means of the expression:

$$w_j = \sqrt{\sum_{i=1}^{n} \exp\left(-D_i/D_0\right)\sqrt{s_i}}, \tag{7}$$

where $w_j$ (unitless) represents the weight for grid cell $j$, $D_i$ (km) the distance to grid cell $i$, $D_0$ (km) the range of influence, and $s_i$ the number of stations for grid cell $i$. The summation is over all grid cells $i = 1, 2, \ldots, n$ within a 1000 km radius of grid cell $j$. $D_0$ was set to 25 km using trial and error. Typical values for $w$ range from 0 in completely ungauged regions to 1 for

grid cells containing only one isolated station and approximately 2.5 in densely gauged regions (e.g., in southeast Australia). As the available gauge observations vary over time, the associated weight maps vary accordingly.

In the fourth and final merging stage (see Figure 1), the long-term mean $P$ was downscaled in a step wise manner using the $P$ anomalies first to the monthly time scale, then to the daily time scale, and finally to the 3-hourly time scale to yield the MSWEP dataset.

## 3  MSWEP performance evaluation

### 3.1  Evaluation using FLUXNET gauge observations

Observed $P$ data from 125 tower sites included in the FLUXNET dataset (Baldocchi et al., 2001; Baldocchi, 2008) were used to evaluate eight $P$ datasets: (i) the final MSWEP, (ii) the gauge-only MSWEP, (iii) the satellite-only MSWEP, (iv) the reanalysis-only MSWEP, (v) WFDEI (adjusted using CRU TS3.1), (vi) GPCP-1DD (version 1.2), (vii) TMPA 3B42 (version 7), and

(viii) CPC Unified (version 1.0 and real time; see Table 1 for details). The FLUXNET data were used for this purpose because they are completely independent; they have not been used in the development of any of the $P$ datasets. The different MSWEP





variants were obtained by setting the weights of the other two sources to zero (e.g., for obtaining the gauge-only MSWEP, the satellite and reanalysis weights were set to zero). All $P$ data with a resolution $< 0.5°$ were resampled to $0.5°$ using averaging for consistency. Only stations with $> 120$ days of data (not necessarily consecutive) during the period 1998–2015 for all datasets were used. The list of stations is provided in the Appendix.

Three performance metrics were considered. The first is the Pearson correlation coefficient ($R$), which ranges from $-1$ to 1, with higher values corresponding to better performance. The second is the root-mean-square error (RMSE), which ranges from 0 to $\infty$, with lower values corresponding to better performance. The third is the absolute bias ($B$), defined as:

$$B = \left| \frac{\overline{P_e} - \overline{P_r}}{\overline{P_e} + \overline{P_r}} \right|, \tag{8}$$

where $\overline{P_e}$ is the long-term mean of the gridded estimates and $\overline{P_r}$ the long-term mean of the station record. Bias values range
from 0 to 1, with lower values corresponding to better performance. The $R$ and RMSE values were computed based on daily (rather than 3-day mean) $P$ data.

## 3.2    Evaluation using hydrological modeling

The performance of the previously mentioned eight $P$ datasets (see Section 3.1) was further evaluated using hydrological modeling for 9011 catchments ($< 50\,000$ km$^2$). Specifically, we calibrated the simple conceptual hydrological model HBV
(Bergström, 1992; Seibert and Vis, 2012) against daily $Q$ observations using the widely used Nash and Sutcliffe (1970) Efficiency (NSE) objective function with $P$ from each of the different datasets. All $P$ data with a resolution $< 0.5°$ were resampled to $0.5°$ using averaging for consistency. The observed daily $Q$ and associated catchment boundary data were obtained from the same three sources as those used for the CHPclim bias correction (see Section 2.1). Catchments were required to have an area $< 50\,000$ km$^2$ and a $Q$ record length $> 1$ year between 1998–2015 (not necessarily consecutive).

HBV operates at a daily time step, has two groundwater stores and one unsaturated-zone store, and uses a triangular weighting function to simulate channel routing delays. The model was chosen because of its flexibility, low computational cost, and successful application under a large variety of climatic and physiographic conditions (e.g., Te Linde et al., 2008; Deelstra et al., 2010; Plesca et al., 2012; Beck et al., 2013; Vetter et al., 2015; Valéry et al., 2014; Beck et al., 2016a). Besides daily $P$ time series, the model requires daily time series of $E_p$ and $T_a$ as inputs. $E_p$ was calculated using the Penman (1948) equation as
given by Shuttleworth (1993) with daily net radiation, $T_a$, atmospheric pressure, wind speed, and relative humidity derived from WFDEI, and surface albedo from monthly climatic data derived from the European Space Agency (ESA) GlobAlbedo dataset (Muller et al., 2011). $T_a$ data were also taken from WFDEI. Only a single "elevation-vegetation zone" (Seibert and Vis, 2012) was considered for each catchment. Table 3 lists the model parameters including their calibration range. The stores were initialized by running the model for the first 10 years of the record if the record length was $\geq 10$ years, or by running the model
multiple times for the entire record if the record length was $< 10$ years.

The model was recalibrated for each $P$ dataset for the period with simultaneous observed $Q$ and input data in a lumped fashion to minimize the computational time. For the calibration we used the ($\mu + \lambda$) evolutionary algorithm implemented using the Distributed Evolutionary Algorithms in Python (DEAP) toolkit (Fortin et al., 2012). The calibration procedure was almost





**Table 3.** HBV model parameter descriptions and calibration ranges.

| Parameter (units) | Description | Minimum | Maximum |
|---|---|---|---|
| TT (°C) | Threshold temperature when $P$ is simulated as snowfall | −2.5 | 2.5 |
| CWH (−) | Water-holding capacity of snow | 0 | 0.2 |
| CFMAX (mm °C$^{-1}$ d$^{-1}$) | Melt rate of the snowpack | 0.5 | 5 |
| CFR (−) | Refreezing coefficient | 0 | 0.1 |
| FC (mm) | Maximum water storage in the unsaturated-zone store | 50 | 700 |
| LP (−) | Soil moisture value above which $E$ reaches $E_{\mathrm{p}}$ | 0.3 | 1 |
| BETA (−) | Shape coefficient of recharge function | 1 | 6 |
| UZL (mm) | Threshold parameter for extra outflow from upper zone | 0 | 100 |
| PERC (mm d$^{-1}$) | Maximum percolation to lower zone | 0 | 6 |
| K0 (d$^{-1}$) | Additional recession coefficient of upper groundwater store | 0.05 | 0.99 |
| K1 (d$^{-1}$) | Recession coefficient of upper groundwater store | 0.01 | 0.8 |
| K2 (d$^{-1}$) | Recession coefficient of lower groundwater store | 0.001 | 0.15 |
| MAXBAS (d) | Length of equilateral triangular weighting function | 1 | 5 |

the same as in Beck et al. (2016b); however, to limit the computational time we reduced the population size ($\mu$) to 12 (from 24), the recombination pool size ($\lambda$) to 24 (from 48), and the number of generations to 10 (from 25).

## 4   Results and discussion

### 4.1   Bias correction of CHPclim

The CHPclim dataset is used to determine the long-term mean of MSWEP. Although CHPclim has been adjusted for orographic effects, it does not explicitly account for wind-induced gauge under-catch and is thus likely to underestimate $P$, particularly in snow-dominated regions. We tested two approaches to bias correct CHPclim. Figure 2a shows bias correction factors computed using country-specific CR equations and a global 5-km map of mean wind speed. The spatial patterns agree reasonably well with those from three previous studies based on interpolation of CR values computed from daily station observations of $P$, $T_{\mathrm{a}}$,

and wind speed (Legates and Willmott, 1990; Adam and Lettenmaier, 2003; Yang et al., 2005). Figure 2b shows bias correction factors inferred for the 13 762 regions using the Zhang et al. (2001) relationship in combination with $Q$ observations and $E_{\mathrm{p}}$ estimates (see Section 2.1 and Equation 3). These values were subsequently interpolated to obtain a continuous global map (see Figure 2c). The resulting bias correction factors are highest in Alaska, northern Canada, the Andes, Scandinavia, the Central Asian mountain ranges, and northeastern Russia (Figure 2c).

There is little agreement between Figures 2a and 2c, particularly in western Russia where the use of $P$ corrected using the country-specific equations to force a hydrological model would probably result in appreciable $Q$ overestimation. Although further research is needed to ascertain the exact cause of this discrepancy, these results suggest firstly that the country-specific CR equations are unsuitable for global application; and secondly, that there is probably still a substantial amount of orographic $P$ unaccounted for in CHPclim. Indeed, almost all regions with high bias correction values ($> 1.25$) in Figure 2c exhibit





a low density of $P$ gauges predominantly located at low elevations in regions with complex topography which might have confounded the estimation of orographic $P$ in CHPclim. The global patterns of Figure 2c agree reasonably well with those from Adam et al. (2006), who estimated the bias in $P$ due to orographic effects at global scale using a similar approach based on $Q$ observations from 357 (mountainous) catchments ($> 10\,000$ km$^2$).

5   Figure 2d presents the CHPclim $P$ adjusted using the interpolated bias correction factors shown in Figure 2c. The bias correction of CHPclim resulted in an increase in global terrestrial mean annual $P$ (Antarctica excluded) from 817 mm yr$^{-1}$ to 877 mm yr$^{-1}$, amounting to an increase of 7.4 %. This number is lower than the 17.9 % increase reported by Adam et al. (2006), possibly because CHPclim has already been corrected for orographic effects. Our estimate is close to the 850 mm yr$^{-1}$ cited by Schneider et al. (2014) based on the GPCC dataset, which includes orographic corrections and accounts for gauge 10   under-catch using Legates and Willmott (1990).

## 4.2   Gauge-based evaluation of gridded $P$ datasets

We evaluated eight non-gauge adjusted gridded $P$ datasets in terms of temporal variability using observations from thousands of gauges around the world, to assess their value for inclusion in MSWEP (see Table 2). All datasets with a resolution $< 0.5°$ were resampled to $0.5°$ for this exercise for consistency. Figure 3 shows 3-day temporal correlation coefficients calculated 15   between rainfall derived from the datasets and the gauges for $0.5°$ grid cells containing at least one gauge. The satellite datasets CMORPH, GSMaP-MVK, and TMPA 3B42RT exhibit very similar performance at global scale (Figure 3a-c). In agreement with previous regional studies (e.g., Brown, 2006; Hirpa et al., 2010; Thiemig et al., 2012; Peña Arancibia et al., 2013; Cattani et al., 2016), PERSIANN performed rather poorly (Figure 3d), possibly because it is predominantly based on cloud-top IR observations. SM2RAIN-ASCAT performed similarly to PERSIANN (Figure 3e). Conversely, Brocca et al. (2014) found 20   SM2RAIN-ASCAT to perform similarly to TMPA 3B42RT based on 5-day correlations. The lower performance obtained here for SM2RAIN-ASCAT could be due to the use of only 4 years of data for SM2RAIN-ASCAT (2007–2010) versus 11 years for TMPA 3B42RT (2000–2010). However, the performance of SM2RAIN-ASCAT appears to be somewhat complementary to the other satellite datasets in that it performs best in the western USA where the other satellite datasets perform rather poorly (Figure 3). Better results could potentially be obtained by applying the SM2RAIN technique to a multi-satellite soil moisture 25   product, for example the Essential Climate Variable Soil Moisture (ECV-SM) dataset (Liu et al., 2011).

The three reanalysis datasets exhibited very similar performance patterns overall, although ERA-Interim consistently performed slightly better than the other reanalysis datasets, in line with several earlier regional studies evaluating different atmospheric variables (e.g., Bromwich et al., 2011; Bracegirdle and Marshall, 2012; Jin-Huan et al., 2014). Although NCEP-CFSR performed reasonably well, it was not incorporated in MSWEP due to the presence of wave-like artifacts around mountain 30   ranges (cf. Seguinot et al., 2014). Patterns of temporal correlations obtained for the USA for the satellite and reanalysis datasets (Figure 3) correspond well with those obtained by Tian et al. (2007) and Ebert et al. (2007). Figure 4 presents the difference in 3-day rainfall correlation coefficients between TMPA 3B42RT and ERA-Interim, highlighting the complementary performance of satellite and reanalysis $P$ datasets reported in previous large-scale studies (Janowiak, 1992; Huffman et al., 1995; Xie and Arkin, 1996, 1997; Adler et al., 2001; Ebert et al., 2007; Serrat-Capdevila et al., 2013; Peña Arancibia et al., 2013; Xie and





**Figure 2.** Global maps of (a) bias correction factors estimated using country-specific equations, (b) bias correction factors inferred using the Zhang et al. (2001) relationship from $Q$ observations and $E_p$ estimates, (c) bias correction factors computed by interpolating the Zhang et al. (2001)-based values, and (d) mean annual $P$ (mm yr$^{-1}$) from CHPclim bias adjusted using the interpolated bias correction factors and used for MSWEP. The data points in (b) represent the centroids of the catchments ($n = 12\,493$) and the interstation regions ($n = 1651$). Note that there are many values in (b) $< 1$, but these are not of interest here and thus are shown as 1.



Joyce, 2014). Based on these results, we decided to use CMORPH, GSMaP-MVK, TMPA 3B42RT, ERA-Interim, and JRA-55 for determining the rainfall variability for ungauged regions in MSWEP.

Figure 5 shows the 3-day temporal correlation between snowfall derived from the $P$ datasets and the gauges for 0.5° grid cells containing at least one gauge. All five satellite datasets consistently exhibit very low correlation coefficients (Figures 5a–e), due to the well-documented failure of retrieval techniques over snow- and ice-covered surfaces, and more generally their inability to detect snowfall in all conditions (e.g., Ferraro et al., 1998; Levizzani et al., 2011; Ebert et al., 2007; Yong et al., 2013; Peng et al., 2014). Conversely, the reanalysis datasets showed high correlation coefficients for most grid cells (Figure 5f–h), even exceeding those obtained for rainfall estimation (Figure 3f–h), possibly because almost all snow originates from (non-convective) large-scale synoptic weather systems which tend to be represented well by atmospheric models. Based on these results, we decided to use ERA-Interim and JRA-55 for determining the snowfall variability for ungauged regions in MSWEP.

Weight maps for MSWEP were calculated for each of the selected $P$ datasets based on the 3-day correlation coefficients. The weight maps for the satellite and reanalysis datasets are static in time, although we used the rainfall weight for grid cells with $T_a > 3°C$ and the snowfall weight for grid cells $T_a < 3°C$. Conversely, the weight maps for the gauges were computed based on gauge density and thus vary daily depending on the available gauge observations. Figure 6a shows the sum of the weights used to merge the gauge, satellite, and reanalysis components of MSWEP during merging stage 3 (see Figure 1) at the daily time scale on an arbitrarily chosen day (April 25, 2006). The relative contributions of the gauge, satellite, and reanalysis components to the total weight are shown in Figures 6b, 6c, and 6d, respectively.

### 4.3 Evaluation of MSWEP using FLUXNET gauge observations

Observed $P$ data from 125 FLUXNET stations were used to evaluate the final MSWEP, the MSWEP gauge, satellite, and reanalysis components, and four existing state-of-the-art gauge-adjusted $P$ datasets (WFDEI-CRU, GPCP-1DD, TMPA 3B42, and CPC Unified). Figure 7 presents box plots of the distribution of $R$, RMSE, and $B$ values obtained for each dataset. The merged MSWEP performed better than the individual MSWEP components in terms of both $R$ and RMSE (Figure 7a and 7b, respectively), confirming the effectiveness of the employed merging scheme. The merged MSWEP also appears to perform better overall than all four existing $P$ datasets in terms of both $R$ and RMSE (Figures 7a and 7b, respectively), providing the best estimates in terms of $R$ for 60.0 % of the stations and in terms of RMSE for 68.8 % of the stations. Interestingly, MSWEP performed average in terms of absolute bias (Figure 7c), obtaining the best $B$ value for only 9.6 % of the stations, which is somewhat unexpected given the sophisticated interpolation techniques employed in producing the CHPclim dataset (Funk et al., 2015b). A possible reason for this is that the FLUXNET $P$ data are generally not corrected for gauge under-catch. Moreover, some of the $P$ records may be subject to measurement bias—at the Chokurdakh station in northeastern Russia (site code RU-Cok; 70.82°N, 147.49°E), for example, snowfall was not measured at all (A. Budishchev, pers. comm.). The relatively good performance of the reanalysis-only MSWEP and WFDEI-CRU datasets (Figures 7a and 7b) confirms the usefulness of ERA-Interim for gap-filling of FLUXNET $P$ records (Vuichard and Papale, 2015).





**Figure 3.** Correlation coefficients calculated between 3-day mean rainfall derived from the datasets and the gauges. Each data point represents a 0.5° grid cell containing at least one gauge.





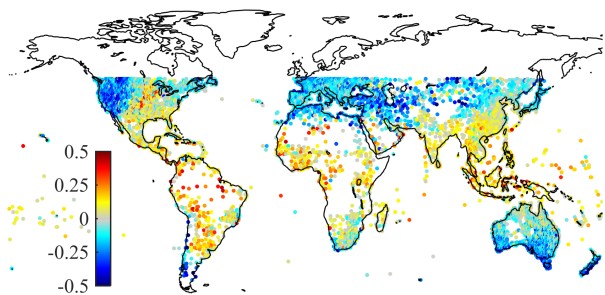

**Figure 4.** The difference in 3-day rainfall correlation coefficients between TMPA 3B42RT and ERA-Interim. Each data point represents a $0.5°$ grid cell containing at least one gauge ($n = 6668$). Red and yellow indicates TMPA 3B42RT performs better, whereas blue indicates ERA-Interim performs better.

## 4.4 Evaluation of MSWEP using hydrological modeling

We further evaluated the performance of MSWEP based on hydrological modeling for 9011 catchments ($< 50\,000$ km$^2$). Figure 8 presents daily NSE scores obtained after the calibration of HBV, in turn with daily $P$ data from the final MSWEP, the gauge-only MSWEP, the satellite-only MSWEP, and the reanalysis-only MSWEP. The respective datasets attained the highest calibration NSE for 28.7 %, 53.1 %, 6.7 %, and 11.5 % of the catchments. The good results for the gauge-only MSWEP are, however, somewhat misleading since the large majority of the catchments are located in regions with very dense $P$ measurement networks (the USA, Europe, and Australia; cf. Figures 6b and 8a). Figure 8b shows median calibration NSE values for $5°$ latitude bands. For most latitudes, the final MSWEP performs either better or only slightly worse than the best performing single-source MSWEP (Figure 8b), confirming the efficacy of the merging scheme. The final MSWEP performs particularly good at latitudes $< 17.5°$N/S and $> 67.5°$N (Figure 8b) where $P$ gauge densities are relatively low. The reanalysis-only MSWEP exhibits markedly lower scores for the 17.5–22.5°N latitude band (Figure 8b), mainly reflecting the low performance achieved for Puerto Rico, perhaps due to the coarse resolution of the atmospheric models used to produce the reanalysis datasets.

To examine in more detail how the final MSWEP, the gauge-only MSWEP, the satellite-only MSWEP, and the reanalysis-only MSWEP perform in regions with dense versus sparse $P$ gauge networks, Figure 9a plots the median calibration NSE for the $P$ datasets as a function of catchment-mean distance to the closest gauge. For the 7953 well-gauged catchments, defined here by catchment-mean distance to the closest $P$ gauge $< 25$ km, the final MSWEP performed slightly worse than the gauge-only MSWEP. However, only 16.1 % of the global land surface (excluding Antarctica) has a $P$ gauge located at $< 25$-km distance. For the 1058 sparsely gauged catchments, defined here by catchment-mean distance to the closest $P$ gauge $> 25$ km, MSWEP performs overall considerably better than the gauge-only MSWEP. Contrary to expectations, the median NSE scores increase with increasing distance to the closest $P$ gauge for several of the $P$ datasets, which is primarily because the more sparsely gauged groups contain less (semi-)arid catchments, which tend to exhibit lower NSE scores.





**Figure 5.** Correlation coefficients calculated between 3-day mean snowfall derived from the datasets and the gauges. Each data point represents a 0.5° grid cell containing at least one gauge.



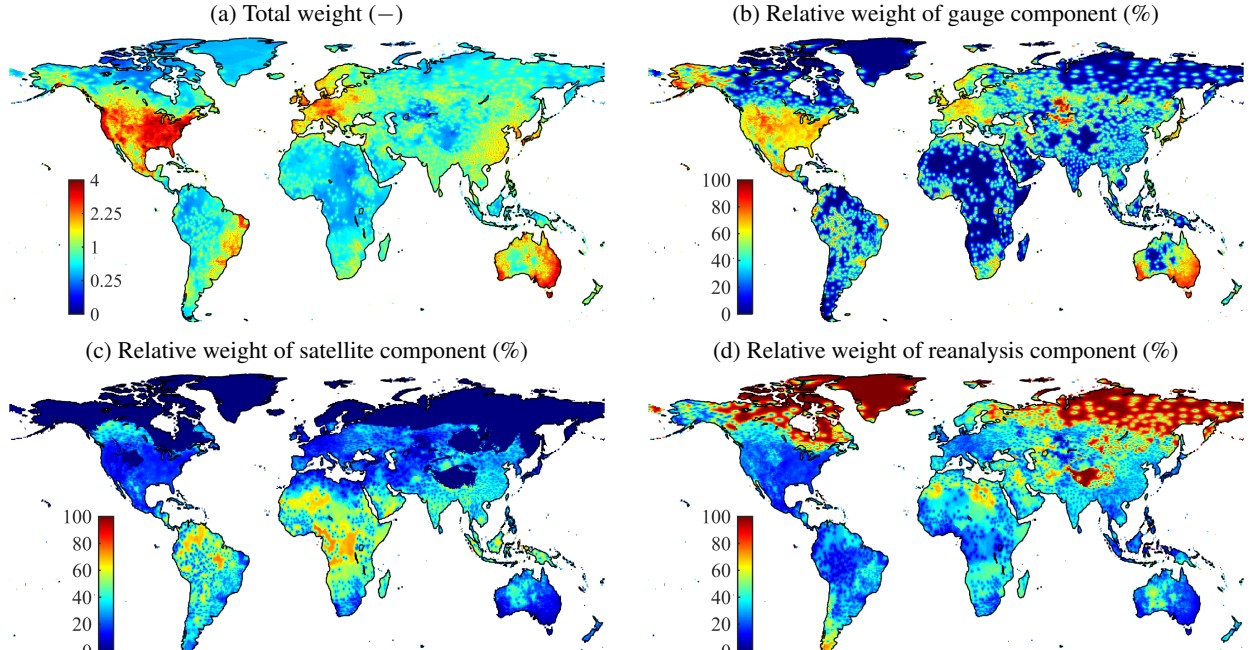

**Figure 6.** Map (a) shows the sum of the gauge weight (see Equation 7) and the satellite and reanalysis weights (derived from 3-day correlation coefficients at surrounding gauges) used during merging stage 3 (see Figure 1) at the daily time scale on an arbitrarily chosen day (April 25, 2006; note the non-linear color scale). The contributions of the gauge, satellite, and reanalysis components to the total weight on the same day are shown in (a), (b), and (c), respectively.

Figure 10 presents daily NSE scores obtained after calibration of HBV in turn with daily $P$ data from MSWEP and four existing state-of-the-art gauge-corrected datasets (WFDEI-CRU, GPCP-1DD, TMPA 3B42, and CPC Unified). MSWEP obtained the highest calibration NSE for 40.4 % of the catchments, while the other datasets obtained the highest calibration NSE for 12.2 %, 4.1 %, 11.3 %, and 32.1 % of the catchments, respectively (noting that the spatial coverage of TMPA 3B42 is limited to $< 50°$N/S). The median calibration NSE score obtained by MSWEP is 0.49 versus 0.30, 0.22, 0.35, and 0.45 for the other datasets, respectively. However, similar to the gauge-only MSWEP (see Figure 8), the scores for CPC Unified are probably somewhat inflated since most catchments are located in regions with very high $P$ gauge densities. Indeed, the picture changes somewhat when we examine median calibration NSE scores for $5°$ latitude bands; MSWEP performs either similar to or substantially better than the other $P$ datasets for all latitudes (Figure 10b). The good performance of WFDEI-CRU for the central Interior Plains of North America (Figure 10a) appears to be attributable to $P$ overestimation by the other datasets including MSWEP.

Figure 9b plots the median calibration NSE for the five $P$ datasets (MSWEP, WFDEI-CRU, GPCP-1DD, TMPA 3B42, and CPC Unified) as a function of catchment-mean distance to the closest gauge. For the 7953 well-gauged catchments (defined by catchment-mean distance to the closest $P$ gauge $< 25$ km), MSWEP obtained a median NSE of 0.49 versus 0.30, 0.21, 0.35, 0.47 for the other $P$ datasets, respectively, with MSWEP obtaining the best NSE in 39.0 % of these catchments. However,





**Figure 7.** Box plots of the distribution of (a) daily correlation ($R$), (b) root-mean-square error (RMSE), and (c) absolute bias ($B$) values computed between observed and estimated $P$ at the 125 FLUXNET sites. Note that for the satellite-only MSWEP only the 113 stations located $< 60°$N/S were used, while for TMPA 3B42 only the 87 stations located $< 50°$N/S were used. The locations of the 125 FLUXNET stations and the $R$ values obtained by MSWEP are shown in (d). The list of stations is provided in the Appendix.



**Figure 8.** The $P$ dataset with the best daily NSE score obtained after calibration of HBV is shown in (a). Also shown are (b) median NSE scores for 5° latitude bands and (c) the number of catchments ($n$) for each 5° latitude band. Each data point in (a) represents a catchment centroid ($N = 9011$).




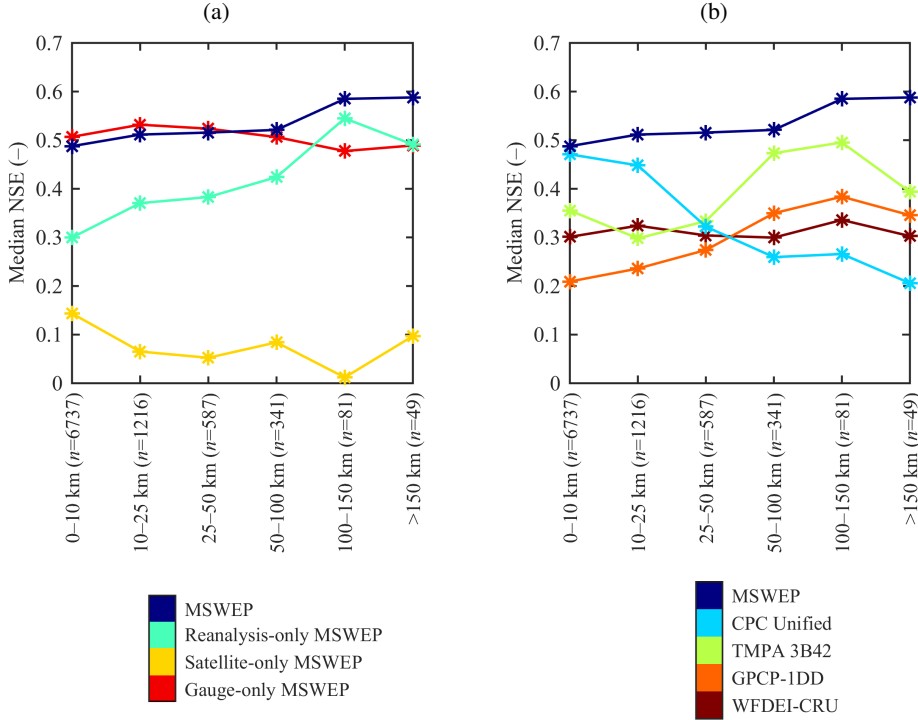

**Figure 9.** The change in median calibration NSE for the $P$ datasets as a function of catchment-mean distance to the closest $P$ gauge. The distance to the closest $P$ gauge was computed using the GPCC and CPC Unified datasets.

as mentioned previously, only 16.1 % of the global land surface (excluding Antarctica) has a $P$ gauge located at $< 25$-km distance. For the 1058 sparsely gauged catchments (defined by catchment-mean distance to the closest $P$ gauge $> 25$ km), MSWEP obtained a median NSE of 0.52 versus 0.30, 0.32, 0.39, 0.29 for the other $P$ datasets, respectively, with MSWEP obtaining the best NSE in 51.6 % of these catchments. Thus, MSWEP performs slightly better than CPC Unified in well-

5 monitored regions, whereas it performs considerably better than all other datasets in sparsely gauged regions which comprise most (83.9 %) of the global land surface.

Figure 11 presents daily NSE scores obtained after calibration of HBV forced with MSWEP. The patterns obtained for the USA exhibit good agreement with those obtained in six previous studies using different hydrological models and forcing data (Lohmann et al., 2004; Xia et al., 2012; Newman et al., 2015; Bock et al., 2015; Essou et al., 2016; Beck et al., 2016b). The

10 NSE values are consistently lower in (semi-)arid regions in North America, Africa, and Australia, likely due to the highly non-linear rainfall-runoff response, the high transmission losses, and the flashy nature of the $Q$ under these conditions (Pilgrim et al., 1988; Ye et al., 1997; Lidén and Harlin, 2000). The lower scores obtained for polar catchments in North America and Iceland are attributable to $P$ underestimation, suggesting that there is still some $P$ unaccounted for. Note that a low score for a particular catchment does not necessarily mean that MSWEP is unreliable as errors in the $Q$, $E_p$, or catchment boundary data

or flow regulation could also be responsible.





**Figure 10.** The $P$ dataset with the best daily NSE score obtained after calibration of HBV is shown in (a). Also shown are (b) median NSE scores for 5° latitude bands and (c) the number of catchments ($n$) for each 5° latitude band. Each data point in (a) represents a catchment centroid ($N = 9011$).



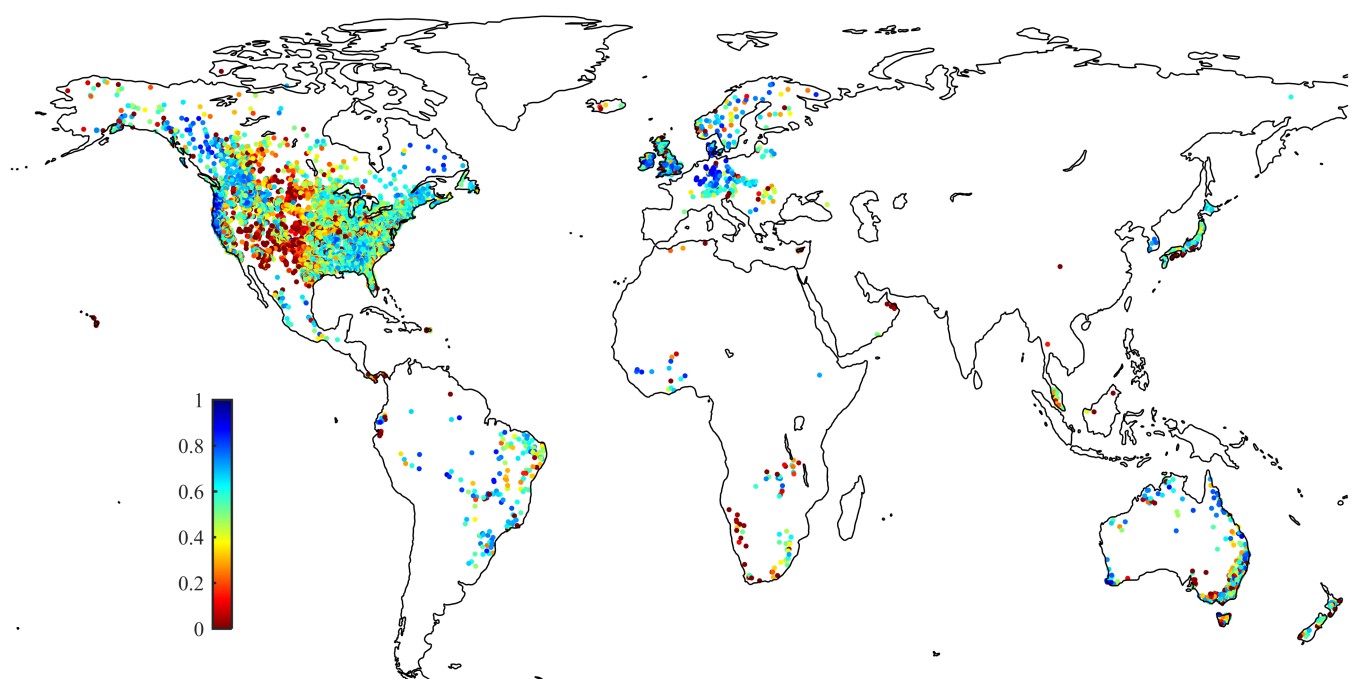

**Figure 11.** Daily NSE scores obtained after calibration of HBV forced by MSWEP. Each data point represents a catchment centroid ($N =$ 9011).

Hydrological modeling has been used in numerous previous studies to evaluate the quality of $P$ datasets (e.g., Su et al., 2008; Collischonn et al., 2008; Voisin et al., 2008; Bitew et al., 2012; Li et al., 2013; Falck et al., 2015; Tang et al., 2016). However, these previous studies were generally regional in nature and used $Q$ observations from a relatively small number of stations (7, 23, 9, 1, 22, 300, and 1, respectively, for the cited studies), which may have led to conclusions with limited global-scale applicability. Moreover, some of these studies did not recalibrate the hydrological model for each $P$ dataset (Su et al., 2008; Voisin et al., 2008; Li et al., 2013), which may have led to additional uncertainty since hydrological models are inevitably imperfect representations of reality and thus produce uncertain estimates even with 'perfect' meteorological forcing. Conversely, we used $Q$ observations from a large set of 9011 stations around the globe spanning all terrestrial latitudes and covering a broad range of physiographic and climatic conditions, which allowed us to examine how $P$-gauge proximity influences the calibration NSE. Furthermore, we recalibrated the model for each catchment and $P$ dataset, thereby minimizing the uncertainty in model parameters.

One could argue that the evaluation of MSWEP using hydrological modeling presented here is not completely independent, since the $Q$ observations used for the evaluation have also been used in the development of MSWEP for the bias correction of CHPclim (see Section 2.1). However, the evaluation is in fact nearly completely independent since the results for a particular catchment will not change much, if at all, if that catchment was excluded from the bias adjustment exercise, since the bias correction factors were computed for each grid cell as the median of the 10 closest catchments. Moreover, the large majority





of catchments are located in regions not affected by the bias correction (81.7 % of the catchments have an interpolated bias correction factor between 1 and 1.02; see Figure 2c). Furthermore, it is noted that all $P$ data were resampled to 0.5° for consistency; both MSWEP and TMPA 3B42 may produce better calibration scores if the original 0.25° resolution data are used.

## 4.5 Caveats and future work

Although MSWEP overall performed better than existing state-of-the-art gauge-adjusted $P$ datasets, some caveats need mentioning:

1. As with many $P$ datasets, spurious temporal discontinuities can occur due to (i) the introduction of the satellite data source in 1998, (ii) satellite sensor degradation and instrument changes, (iii) differences in time coverage among stations, and (iv) changes in the observations assimilated in the reanalysis datasets (e.g., Kang and Ahn, 2015). Thus, as with any $P$ dataset, care should be exercised when using MSWEP for trend analysis.

2. Sub-daily temporal variability in tropical regions should be interpreted with caution prior to the launch of TRMM in 1998, since atmospheric models poorly represent the diurnal cycle in (sub-)tropical regions (Kidd et al., 2013).

3. The estimated bias correction factors are influenced by uncertainties in the $Q$, $E_p$, and catchment boundary data and particularly the interpolation. The spatial bias correction factors are available along with the data, so the original, non-corrected long-term averages can be reconstructed.

4. The use of PRISM data for the conterminous USA and the restriction of the bias correction to regions with snowfall and/or complex topography may result in spatial discontinuities in the long-term average of MSWEP.

5. Although MSWEP has a 0.25° spatial resolution (corresponding to ∼25 km at the equator), the effective spatial resolution reflects the gauge density in regions where the gauge component dominates (Figure 6b), while it reflects the resolution of the atmospheric models (∼80–125 km) in regions where the reanalysis component dominates (Figure 6d). In regions where the satellite component dominates the effective resolution can be assumed to be 0.25° (Figure 6c).

Future updates and improvements of MSWEP are intended to take advantage of new and improved data sources. For example, MSWEP may be updated with ERA5, the planned successor to ERA-Interim, while the TMPA dataset may be replaced by the Integrated Multi-satellitE Retrievals for GPM (IMERG) dataset (Huffman et al., 2014) once it has been extended backwards to the start of the Tropical Rainfall Measuring Mission (TRMM) era. Given the promising complementary information from SM2RAIN-ASCAT (Figure 3), any such dataset ingesting soil moisture retrievals from multiple satellites should also be considered for inclusion in MSWEP, if available. Moreover, the gauge component of MSWEP may be improved by accounting for the variable correlation length in the interpolation (cf. Funk et al., 2015a), while the biases at high northern latitudes may be improved using Gravity Recovery and Climate Experiment (GRACE) data (cf. Behrangi et al., 2016; Wang et al., 2016). Finally, we hope to explore the possibility of producing a near-real time variant of MSWEP. Many of the datasets incorporated in MSWEP are available in near-real time, suggesting this is feasible.



## 5  Conclusions

We developed a global gridded $P$ dataset (1979–2015) with several novel features: (i) it takes advantage of a wide range of data sources, including gauge, satellite, and reanalysis data, to provide reliable $P$ estimates at global scale; (ii) it accounts for gauge under-catch and orographic effects using $Q$ observations from an unprecedentedly large set of 13 762 stations worldwide; and (iii) it has a high 3-hourly temporal and 0.25° spatial resolution. The $P$ estimates and ancillary data are available via http://www.gloh2o.org. We summarize our findings as follows:

1. The high-resolution global CHPclim climatic $P$ dataset, which explicitly accounts for orographic effects, still appears to contain widespread $P$ biases. These biases could not be satisfactorily corrected using country-specific equations that account for gauge under-catch, suggesting that CHPclim still underestimates orographic $P$ in some regions, probably mainly due to the topographic bias in gauge placement. We used the Zhang et al. (2001) relationship in combination with $Q$ observations from 13 762 stations around the globe to produce a bias adjusted variant of CHPclim for use in MSWEP.

2. Eight non-gauge adjusted $P$ datasets were evaluated in terms of 3-day temporal variability using observed $P$ data from thousands of gauges around the globe to assess their value for inclusion in MSWEP. For rainfall, CMORPH, GSMaP-MVK, and TMPA 3B42RT performed equally well, while PERSIANN as well as SM2RAIN-ASCAT performed less well. The satellite datasets consistently performed better in the tropics, while the reanalysis datasets consistently performed better at mid to high latitudes. For snowfall, all satellite datasets performed poorly, while the reanalysis datasets performed even better than for rainfall. The obtained correlation coefficients were interpolated to produce global weight maps for the merging scheme of MSWEP.

3. Independent observed $P$ data from 125 FLUXNET tower stations around the globe were used to evaluate the performance of MSWEP as well as four existing state-of-the-art global $P$ datasets (GPCP-1DD, WFDEI-CRU, TMPA 3B42, and CPC Unified). The median $R$ value obtained by MSWEP was 0.67 versus 0.44–0.59 for the other $P$ datasets, with MSWEP obtaining the best $R$ score for 60.0 % of the stations. MSWEP also performed best in terms of RMSE, but performed average in terms of absolute bias.

4. We further evaluated the performance of MSWEP using hydrological modeling for 9011 catchments ($< 50\,000$ km$^2$) around the globe. Specifically, we calibrated the HBV hydrological model against daily $Q$ observations in turn with $P$ from MSWEP and the four previously mentioned existing $P$ datasets. For the 1058 sparsely gauged catchments (defined by catchment-mean distance to the closest $P$ gauge $> 25$ km), representative of 83.9 % of the global land surface (excluding Antarctica), MSWEP obtained a median NSE of 0.52 versus 0.29–0.39 for the other $P$ datasets.

## Appendix:  FLUXNET stations used in this study

The 125 FLUXNET sites used in this study including the primary reference, where available, are: AR-SLu, AR-Vir, AT-Neu, AU-ASM (Cleverly et al., 2013), AU-Ade, AU-Cpr, AU-Cum, AU-DaP, AU-DaS, AU-Dry, AU-Emr, AU-Fog, AU-RDF, AU-





Rig, AU-Tum, AU-Whr, BE-Bra, BE-Lon, BE-Vie, BR-Sa3, CA-Gro (McCaughey et al., 2006), CA-NS1 (Goulden et al., 2006), CA-NS3 (Goulden et al., 2006), CA-NS4, CA-NS5 (Wang et al., 2002), CA-NS6 (Bond-Lamberty et al., 2004), CA-NS7 (Bond-Lamberty et al., 2004), CA-Qfo (Chen et al., 2006), CA-SF1 (Amiro et al., 2006), CA-SF2 (Amiro, 2009), CA-SF3 (Amiro, 2009), CA-TP2 (Arain and Restrepo-Coupe, 2005), CG-Tch, CH-Cha (Zeeman et al., 2010), CH-Fru (Zeeman et al., 2010), CH-Oe1 (Zeeman et al., 2010), CN-Cha, CN-Cng, CN-Dan, CN-Din, CN-Du2, CN-Du3, CN-Ha2, CN-HaM, CN-Qia, CN-Sw2, CZ-BK1, CZ-BK2, DE-Akm, DE-Gri, DE-Hai, DE-Kli, DE-Lkb, DE-Obe, DE-RuS, DE-Spw, DE-Tha, DK-Eng, DK-NuF (Westergaard-Nielsen et al., 2013), DK-Sor, DK-ZaH (Lund et al., 2012), ES-LJu, ES-LgS, ES-Ln2, FI-Hyy, FI-Jok, FR-Gri, FR-Pue, GH-Ank, IT-CA1, IT-CA2, IT-CA3, IT-La2, IT-Lav, IT-PT1, IT-Ren, IT-Ro1, IT-Ro2, IT-Tor, JP-MBF, JP-SMF, MY-PSO, NL-Hor, NL-Loo, NO-Adv, PA-SPn, PA-SPs, RU-Che, RU-Cok, RU-Fyo, RU-Ha1, RU-Sam, RU-SkP, RU-Vrk, SD-Dem (Ardö et al., 2008), SE-St1, US-AR1, US-AR2, US-ARM (Fischer et al., 2007), US-Blo (Lunden et al., 2006), US-CRT (Chu et al., 2014), US-Goo, US-Ha1 (Goulden et al., 1996), US-IB2 (Jastrow, 1997), US-Ivo (Epstein et al., 2004), US-Lin, US-Los (Bakwin et al., 2004), US-MMS (Schmid et al., 2000), US-Me6 (Ruehr et al., 2012), US-Myb, US-Ne1 (Simbahan et al., 2006), US-Ne2 (Amos et al., 2005), US-Ne3 (Verma et al., 2005), US-Oho (Noormets et al., 2008), US-SRM (Scott et al., 2009), US-Syv (Desai et al., 2005), US-Ton (Chen et al., 2007), US-UMd (Nave et al., 2011), US-Var (Ma et al., 2007), US-WCr (Cook et al., 2004), US-WPT (Chu et al., 2014), US-Whs (Scott, 2010), and US-Wkg (Scott et al., 2010), ZA-Kru, and ZM-Mon.

*Acknowledgements.* The MSWEP dataset is freely available via http://www.gloh2o.org. We gratefully acknowledge the $P$ dataset developers for producing and making their datasets available. The GRDC and the USGS are thanked for providing most of the observed $Q$ data. We further wish to thank Vera Thiemig, Feyera Aga Hirpa, and Jutta Thielen-del Pozo for fruitful discussions on the intercomparison and evaluation of $P$ datasets; Roxana Petrescu and Artem Budishchev for useful comments regarding the FLUXNET dataset; Alessandro Cescatti for suggestions on an earlier draft; and Emanuel Dutra for advice on the merging procedure. This work used eddy covariance data acquired and shared by the FLUXNET community, including these networks: AmeriFlux, AfriFlux, AsiaFlux, CarboAfrica, CarboEuropeIP, CarboItaly, CarboMont, ChinaFlux, Fluxnet-Canada, GreenGrass, ICOS, KoFlux, LBA, NECC, OzFlux-TERN, TCOS-Siberia, USCCC, and Swiss FluxNet. The FLUXNET eddy covariance data processing and harmonization was carried out by the ICOS Ecosystem Thematic Center, AmeriFlux Management Project and Fluxdata project of FLUXNET, with the support of CDIAC, and the OzFlux, ChinaFlux and AsiaFlux offices. This research recseived funding from the European Union Seventh Framework Programme (FP7/2007–2013) under grant agreement no. 603608, "Global Earth Observation for integrated water resource assessment": eartH2Observe. Vincenzo Levizzani acknowledges partial funding from the "Progetto di Interesse NextData" of the Italian Ministry of Education, University, and Research (MIUR). Diego Miralles acknowledges financial support from The Netherlands Organization for Scientific Research through grant no. 863.14.004. The views expressed herein are those of the authors and do not necessarily reflect those of the European Commission.





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
