# Peer review of "MSWEP: 3-hourly 0.25° global gridded precipitation (1979–2015) by merging gauge, satellite, and reanalysis data"

_Hydrology and Earth System Sciences, 2016_

## Referee Comment (RC1) · Anonymous Referee #1 · 21 Jun 2016

In the manuscript titled "MSWEP: 3-hourly 0.25 global gridded precipitation (1979–2015) by merging gauge, satellite, and reanalysis data" authors have merged several satellite and reanalysis only precipitation products. This merged product is later validated using precipitation data sets that are not used in the merging process and using HBV hydrological model outputs. Results of both validation efforts show the merged product is on average superior to input products. The idea of merging different products to obtain a better one sounds trivial, yet in this case it results in a product that may have large application areas. The topic is relevant to HESS Journal, and both the methodology and the validation efforts sound good. I recommend the study to be published after correction of some points.

[Figure]

1.a) I found the methodology section related with HBV model rather short, it would be idea if it is expanded. I guess NSE is calculated between observed and simulated Q values, but I couldn't find this info written explicitly. I am not very familiar with the HBV model, so the parameter calibration part seems not clear to me (e.g., how did authors implemented the calibration, using a particular software? Running the model with different combinations of parameters sampled randomly from their defined range in Table 3?)

1.b) Did HBV calibration and validation efforts use the same runoff (Q) data? If they are the same, then it is very likely that the calibration might fit Q observations too closely (which is a particular advantage on MSWEP compared to other products).

2) Do Reanalysis data have 5m wind dataset instead of converting 80m height to 5m using some wind-profile relation?

3) E = P – Q. If E is only evaporation (line 6, page 7), then what happens to transpiration component?

4) Why normalize absolute bias? The unit of the bias is very important as well.. It would be complementary with RMSE (i.e., the random components can be calculated if non-normalized bias and RMSE are known).

5) Figure 6a, Long-term average weights would have been more meaningful rather than arbitrarily chosen single day.

6) It is not really clear to me why reanalysis HBV performance is much worse than MSWEP given there is only minor difference between them in terms of accuracy of P (Figure 7)?

7) NSE increases with increasing distance for Reanalysis? How come farther away gauges give more reliable precipitation information compared to closer gauges? I might be missing something simple.

Minor

- Page 3, line 8, "These datasets have . . .".

- Table 1, CMORPH does not use gauge data, why it is included in "Gauge, satellite" row? There is another row specifically dedicated to "satellite" (products 19 and 20).

- Consider using the word "using" instead of "in turn"/"in turn with". It is very confusing.

- Figs. 8-11 captions should include very brief info about the parameter used in NSE calculation (i.e., Q).

[Figure]

---

## Referee Comment (RC2) · N. Wanders (Referee) · 29 Jun 2016

The manuscript describes a newly developed global precipitation data, that takes advantage of existing products and uses a weighing approach to merge that into on consistent product. The manuscript is well written and I think the authors did a great job in creating this new MSWEP product. Nonetheless, I have some comments that I think should be thought about and or addressed in a revised version of the manuscript before publication.

Major comments

Figure 1, If I understand correctly, the authors use the monthly performance weights

for the daily and 3-hourly merging? It is unclear if the authors first compute 3-day precipitation timeseries and then compute 1 annual correlation values between those 3-day precipitation values and the observations or that the 3-day precipitation timeseries are used to compute an annual estimate of the correlations which is dependent on the day of the year (i.e. varies throughout the year).

In addition to the previous comment, if annual correlations are used or monthly weights are computed as indicated in Figure 1 this will have an impact on the daily merging. I can imagine that some products better capture the 3-hourly variability than the 3-day precipitation average. For example, some of the latest satellite precipitation products might be in this latter category, while I think a product like WFDEI, does a better job in monthly totals. A product like WFDEI is more heavily bias corrected and other observations are assimilated into the product (e.g. soil moisture). Therefore, WFDEI will get a high weight from the monthly analysis, while on the 3-hourly resolution the performance might not be as good as for a satellite product.

Why do the authors perform a product validation at 0.5 degree? I understand that the gauge density might be low at 0.25 and that more erroneous observations might be included, however, that is the resolution that product is going to be used at by users. I think including a validation at that resolution might prove valuable, if not in the main manuscript, maybe in a supplement.

Why did the authors select this merging procedure and not a more standard Bayesian methods where the errors between the products are weighted and their cross-correlation is taken into account. I might have missed it, but reading the manuscript, I come to the impression that the cross-correlation in the errors between the different products is not taken into account. I think that a large part of the errors in most of the product show a strong cross-correlation, which could be exploited in the merging of the products. That could further strengthen the added value of the MSWEP product compared to the existing data.

One thing that I missed after reading the manuscript is the development of an uncertainty product. The authors have P anomalies from all products and they have the weights, so they can indicate an uncertainty on their product. This would significantly strengthen the MSWEP product, especially in terms of ensemble modelling. Many studies just make assumptions on the uncertainty of a precipitation dataset when they use them for their modelling studies, while the authors are in the position to actually provide this valuable information to the reader and data user. I understand this might be some undertaking and too much for this manuscript, but I think it could be a valuable addition in the future.

Minor comments

Table 1 PRISM is missing from the table

I feel it would be good to elaborate a bit more on the merging in the manuscript, the assumptions made here are very important for the final product. Why is this method chosen over others etc.

How is the performance of the product over mountainous regions, and more specific the Hindu Kush – Himalaya region? Immerzeel et al. (2015) showed in a recent study that most the annual totals of a selection precipitation datasets does not even match the annual discharge ($Q > P$), which indicates some severe biases in the products. Is it possible with MSWEP to correct for these biases or would MSWEP suffer from the same problems? No HBV validation has been done in this region, while it is a region of major importance with regard to water demand, availability etc. Why not perform a quick check to see if $Q < P$ (long-term average to excluding changes in storage) for most of the GRDC stations and see if the annual totals could at least account for the observed discharge. For some of the original products, this would definitely not be the case. This makes me curious to see if MSWEP can overcome that problem.

Reference: Immerzeel, W. W., Wanders, N., Lutz, A. F., Shea, J. M., and Bierkens, M. F. P.: Reconciling high-altitude precipitation in the upper Indus basin with glacier mass

balances and runoff, Hydrol. Earth Syst. Sci., 19, 4673-4687, doi:10.5194/hess-19-4673-2015, 2015.

---

## Short Comment (SC1) · 22 Jul 2016

**Short Comments**

I concur with previous reviewers that the paper is well written and highly of interest for the hydrological community. I believe that the idea of merging satellite-based, gauge-based and reanalysis precipitation data for taking advantage of the benefits of each product is good, and surprisingly not largely investigated so far. I have only two short comments that, in my opinion, should be addressed.

1) The gauge-based evaluation of precipitation datasets is carried out by using the GHCN –D database. I am fully aware that it is difficult to determine the quality of precipitation datasets, but I have not seen in the paper any comments about the reliability and accuracy of the gauge-based dataset. For instance, the spatial representativeness of point measurements might be low even when the average of multiple stations is done. This is particularly problematic at 0.5° resolution. For instance, if satellite datasets were compared with gridded-based dataset (just for checking, not for the merging procedure), what are the results? I believe that a different picture can be obtained (but I could be totally wrong). Can the authors add some additional discussions on this aspect?

2) As hydrologist, I was very interested from the analysis for evaluating MSWEP dataset through hydrological modelling. Particularly, Figure 9a highlights that the overall performance of satellite-only MSWEP dataset is significantly lower than the reanalysis-only and gauge-only MSWEP dataset. Do the authors have an explanation for these large differences? In Figure 8b, satellite-only performance are better than the reanalysis in the tropical region, but (nearly) always lower than the gauge-based product. As the paper represents also one of the first studies performing a comprehensive assessment, on a global scale, of the three sources of precipitation (satellite, reanalysis, raingauge), it would be interesting to extend the discussion of the obtained results through hydrological modelling. I believe it would be highly of interest to the HESS readership.

---

## Author Comment (AC1) · 15 Aug 2016

**Discussion comment 1**

In the manuscript titled "MSWEP: 3-hourly 0.25 global gridded precipitation (1979–2015) by merging gauge, satellite, and reanalysis data" authors have merged several satellite and reanalysis only precipitation products. This merged product is later validated using precipitation data sets that are not used in the merging process and using HBV hydrological model outputs. Results of both validation efforts show the merged product is on average superior to input products. The idea of merging different products to obtain a better one sounds trivial, yet in this case it results in a product that may have large application areas. The topic is relevant to HESS Journal, and both the methodology and the validation efforts sound good. I recommend the study to be published after correction of some points.

We would like to sincerely thank the reviewer for their positive remarks and thorough review.

1.a) I found the methodology section related with HBV model rather short, it would be idea if it is expanded. I guess NSE is calculated between observed and simulated Q values, but I couldn't find this info written explicitly. I am not very familiar with the HBV model, so the parameter calibration part seems not clear to me (e.g., how did authors implemented the calibration, using a particular software? Running the model with different combinations of parameters sampled randomly from their defined range in Table 3?)

We appreciate the suggestion. We now explicitly mention in the methodology and in the captions of Figure 8–10 that the NSE is calculated using simulated and observed $Q$ time series. The discussion m/s already explicitly mentions the software with which we implemented the calibration including a citation (the Distributed Evolutionary Algorithms in Python–DEAP–toolkit; Fortin et al., 2012; see page 11 lines 32–33). The discussion m/s also already explicitly mentions the parameter optimization scheme which we employed on page 11 line 32 (the $(\mu+\lambda)$ evolutionary algorithm). We added an additional reference for readers interested in more details (Ashlock, 2010). Note that the software and optimization scheme can be used with any hydrological model—it is not part of the HBV model or provided with it.

Ashlock, D. (2010): Evolutionary computation for modeling and optimization, Springer Publishing Company, pp. 572.

1.b) Did HBV calibration and validation efforts use the same runoff (Q) data? If they are the same, then it is very likely that the calibration might fit Q observations too closely (which is a particular advantage on MSWEP compared to other products).

HBV was re-calibrated for each precipitation product independently, thus MSWEP was not given any unfair advantage. There was in fact no validation exercise—our objective was to quantify the information content of each product and we are therefore only interested in the calibration NSE scores, which we report in the m/s.

2) Do Reanalysis data have 5m wind dataset instead of converting 80m height to 5m using some wind-profile relation?

Unfortunately, the Vaisala data are only available at 80-m height since they are originally intended for wind-power related applications. Although other reanalysis datasets exist that do provide wind speed estimates at more appropriate heights, they tend to have a much lower spatial resolution and consequently fail to provide realistic wind speed estimates in many mountainous regions.

3) E = P – Q. If E is only evaporation (line 6, page 7), then what happens to transpiration component?

Our definition of evaporation includes transpiration (please see Savenije, 2004, for a discussion on "evaporation" versus "evapotranspiration").

Savenije, H. H. G. (2004), The importance of interception and why we should delete the term evapotranspiration from our vocabulary. *Hydrol. Process.*, 18: 1507–1511. doi: 10.1002/hyp.5563.

4) Why normalize absolute bias? The unit of the bias is very important as well.. It would be complementary with RMSE (i.e., the random components can be calculated if non-normalized bias and RMSE are known).

We did not normalize the bias (*B*). Perhaps the reviewer is asking why we took the absolute value of *B*? This is because, in this study, we are not interested in whether the products generally overestimate or underestimate. Rather, we are interested in how far off the values are on average, or in other words, how well the product performs on average. RMSE computes the difference for each time step and is thus completely unrelated to *B*, which averages each time series prior to calculating the difference.

5) Figure 6a, Long-term average weights would have been more meaningful rather than arbitrarily chosen single day.

We appreciate the suggestion, and have given this some thought, but in the end we decided to only show an example for a single day. This is because, if we would show the long-term average, we would have to do this separately for the pre- and the post-TRMM eras, and for the monthly, daily, and 3-hourly time scales. This would result in 18 figures spanning two pages, which would detract from the main message of the m/s. However, based on this comment we have decided to release the weight maps as part of the NetCDF files so people can use them in their research and derive any spatial-temporal average they like .

6) It is not really clear to me why reanalysis HBV performance is much worse than MSWEP given there is only minor difference between them in terms of accuracy of P (Figure 7)?

The distribution of the FLUXNET stations is completely different from that of the streamflow gauges, leading to very different performance scores. Notably, there is a lack of FLUXNET stations in the tropics, where reanalysis-based products tend to perform poorly.

7) NSE increases with increasing distance for Reanalysis? How come farther away gauges give more reliable precipitation information compared to closer gauges? I might be missing something simple.

Good question. This is due to the uneven distribution of the streamflow gauges around the globe, as explained in the discussion m/s (page 17 lines 20–22): "Contrary to expectations, the median NSE scores increase with increasing distance to the closest P gauge for several of the P datasets, which is primarily because the more sparsely gauged groups contain less (semi-)arid catchments, which tend to exhibit lower NSE scores."

Minor
- Page 3, line 8, "These datasets have . . .".

Thanks for the suggestion. However, we are referring to precipitation products in general, not specifically to the aforementioned products. We therefore prefer to keep "The datasets have …".

- Table 1, CMORPH does not use gauge data, why it is included in "Gauge, satellite" row? There is another row specifically dedicated to "satellite" (products 19 and 20).

As stated in the caption of Table 1, we list for each dataset only the "best" variant. So for CMORPH we refer to the variant that incorporates gauge data (which does in fact exist).

- Consider using the word "using" instead of "in turn"/"in turn with". It is very confusing.

Changed. Thank you for the suggestion.

- Figs. 8-11 captions should include very brief info about the parameter used in NSE calculation (i.e., Q).

We have added to each figure's caption that "The NSE scores have been computed between simulated and observed $Q$ time series."

---

## Author Comment (AC2) · 15 Aug 2016

**Discussion comment 2**

The manuscript describes a newly developed global precipitation data, that takes advantage of existing products and uses a weighing approach to merge that into on consistent product. The manuscript is well written and I think the authors did a great job in creating this new MSWEP product. Nonetheless, I have some comments that I think should be thought about and or addressed in a revised version of the manuscript before publication.

We thank Dr. Wanders for his positive and useful remarks on the m/s.

Major comments
Figure 1, If I understand correctly, the authors use the monthly performance weights for the daily and 3-hourly merging? It is unclear if the authors first compute 3-day precipitation timeseries and then compute 1 annual correlation values between those 3-day precipitation values and the observations or that the 3-day precipitation timeseries are used to compute an annual estimate of the correlations which is dependent on the day of the year (i.e. varies throughout the year).

This may be a misunderstanding. Figure 1 states that, for the daily time scale, we used the same weights as those used at the monthly time scale. Figure 1 also states that the weights used at the monthly time scale are based on 3-day correlations obtained at surrounding gauges. Thus, 3-day correlations are used for the merging at the monthly, daily, as well as 3-hourly time scales. The weights are constant in time, with two exceptions: (1) the satellite weight is zero when the air temperature is <1°C; and (2) the gauge weight depends on the gauge density which varies from day to day. These details can be found in Section 2.3.

In addition to the previous comment, if annual correlations are used or monthly weights are computed as indicated in Figure 1 this will have an impact on the daily merging. I can imagine that some products better capture the 3-hourly variability than the 3-day precipitation average. For example, some of the latest satellite precipitation products might be in this latter category, while I think a product like WFDEI, does a better job in monthly totals. A product like WFDEI is more heavily bias corrected and other observations are assimilated into the product (e.g. soil moisture). Therefore, WFDEI will get a high weight from the monthly analysis, while on the 3-hourly resolution the performance might not be as good as for a satellite product.

In including Figure 1 we did not mean to suggest that we use annual correlations or monthly weights. Furthermore, although WFDEI is one of the five datasets included in the comparative performance evaluation, it is not incorporated in MSWEP and thus has no actual weight assigned to it.

Why do the authors perform a product validation at 0.5 degree? I understand that the gauge density might be low at 0.25 and that more erroneous observations might be included, however,

that is the resolution that product is going to be used at by users. I think including a validation at that resolution might prove valuable, if not in the main manuscript, maybe in a supplement.

We thank the reviewer for this question. The main reason is that not all products are available at 0.25°, and we wanted to rule out resolution differences as a potential cause for the performance differences. However, motivated by the reviewers comment we re-did the analysis using the native resolution of each product. The results were virtually identical however, and we decided not to present them in the m/s.

Why did the authors select this merging procedure and not a more standard Bayesian methods where the errors between the products are weighted and their crosscorrelation is taken into account. I might have missed it, but reading the manuscript, I come to the impression that the cross-correlation in the errors between the different products is not taken into account. I think that a large part of the errors in most of the product show a strong cross-correlation, which could be exploited in the merging of the products. That could further strengthen the added value of the MSWEP product compared to the existing data.

The employed merging method is essentially just a special case of the Bayesian average with the constant $C$ set to 0 (see https://en.wikipedia.org/wiki/Bayesian_average). We do in fact take cross-correlations into account: We assumed that most of the cross-correlation exists among datasets of the same type (satellite or reanalysis), and to account for this cross-correlation we introduced Merging stage 2, in which we first compute a weighted average over all satellite products and over all reanalysis products separately, prior to merging the gauge, satellite, and reanalysis components in Merging stage 3 (see Figure 1).

One thing that I missed after reading the manuscript is the development of an uncertainty product. The authors have P anomalies from all products and they have the weights, so they can indicate an uncertainty on their product. This would significantly strengthen the MSWEP product, especially in terms of ensemble modelling. Many studies just make assumptions on the uncertainty of a precipitation dataset when they use them for their modelling studies, while the authors are in the position to actually provide this valuable information to the reader and data user. I understand this might be some undertaking and too much for this manuscript, but I think it could be a valuable addition in the future.

We agree and intend to explore the estimation of uncertainty in the near future. As the Reviewer correctly states, this is a major endeavor and we are currently pursuing it. However, we want to emphasize that this exercise is not as straightforward as it may seem at first, because for most grid cells there are (1) a limited number of independent estimates and (2) large differences in weights among the estimates. An ungauged arctic grid cell, for example, only has two estimates with any weight assigned to them (JRA-55 and ERA-Interim), which is insufficient to reliably quantify the uncertainty. On the other hand, grid cells containing one or more gauges would have at least three estimates. However, the gauged estimate would have a considerably greater weight, confounding the uncertainty quantification. Although in the tropics there are usually

three estimates (CMORPH, GSMaP, and TMPA 3B42RT) with comparable weights, they are not completely independent and hence would lead to underestimation of the uncertainty.

Minor comments
Table 1 PRISM is missing from the table

Table 1 only lists "(quasi-)global gridded *P* datasets". PRISM is a regional climatic dataset and has therefore been intentionally left out.

I feel it would be good to elaborate a bit more on the merging in the manuscript, the assumptions made here are very important for the final product. Why is this method chosen over others etc.

We agree and have added the following to Section 2: "This method was used since it: (i) is relatively easy to understand and implement; (ii) accommodates the inclusion of datasets with 3-hourly, daily, as well as monthly temporal resolutions; (iii) is largely data-driven (i.e., the weights are based on gauge observations); (iv) accounts for cross-correlation among datasets of the same type (satellite or reanalysis); (v) treats random (i.e., temporally variable) and systematic (i.e., long-term) errors separately; (vi) accounts for gauge under-catch and orographic bias; and (vii) yields reliable estimates (as the comparative performance evaluation described in Section 3 will demonstrate)."

How is the performance of the product over mountainous regions, and more specific the Hindu Kush – Himalaya region? Immerzeel et al. (2015) showed in a recent study that most the annual totals of a selection precipitation datasets does not even match the annual discharge (Q > P), which indicates some severe biases in the products. Is it possible with MSWEP to correct for these biases or would MSWEP suffer from the same problems? No HBV validation has been done in this region, while it is a region of major importance with regard to water demand, availability etc. Why not perform a quick check to see if Q < P (long-term average to excluding changes in storage) for most of the GRDC stations and see if the annual totals could at least account for the observed discharge. For some of the original products, this would definitely not be the case. This makes me curious to see if MSWEP can overcome that problem.

Reference: Immerzeel, W. W., Wanders, N., Lutz, A. F., Shea, J. M., and Bierkens, M. F. P.: Reconciling high-altitude precipitation in the upper Indus basin with glacier mass balances and runoff, Hydrol. Earth Syst. Sci., 19, 4673-4687, doi:10.5194/hess-19-4673-2015, 2015.

MSWEP has indeed been corrected for biases in the Hindu Kush – Himalaya region as well in many other regions with even more severe biases. Figure 2c presents the global map of bias correction factors based on *Q* observations, showing that the bias correction factors for the Hindu Kush – Himalaya region exceed two for most of the area. Thus, we are confident that MSWEP performs better than other *P* products in this regard.

---

## Author Comment (AC3) · 15 Aug 2016

**Discussion comment 3**

I concur with previous reviewers that the paper is well written and highly of interest for the hydrological community. I believe that the idea of merging satellite-based, gaugebased and reanalysis precipitation data for taking advantage of the benefits of each product is good, and surprisingly not largely investigated so far. I have only two short comments that, in my opinion, should be addressed.

We sincerely thank Dr. Brocca for his remarks, to which we respond below.

1) The gauge-based evaluation of precipitation datasets is carried out by using the GHCN –D database. I am fully aware that it is difficult to determine the quality of precipitation datasets, but I have not seen in the paper any comments about the reliability and accuracy of the gauge-based dataset.

It is hard to make any generalizations about the quality of gauge observations since the "true" precipitation is not known. Nevertheless, GHCN-D data have been subjected to stringent quality control procedures (see https://www.ncdc.noaa.gov/oa/climate/ghcn-daily/index.php?name=quality), so we expect the data to be as reliable as possible.

For instance, the spatial representativeness of point measurements might be low even when the average of multiple stations is done. This is particularly problematic at $0.5°$ resolution.

There is indeed a scale discrepancy between the point scale of gauge observations and the grid-cell scale of the datasets, but the quantification of the implications of this discrepancy for our results is confounded by the lack of knowledge on the "true" spatio-temporal precipitation pattern. While approaches have been proposed to correct for this scale discrepancy (see, e.g., Pegram and Bardossy, 2013) they inherently involve many assumptions and uncertainties. Moreover, if we had applied such corrections, it is unlikely the weight maps would have changed much, if at all, as they are based on correlation coefficients (i.e., they are based solely on the temporal dynamics of the data).

Pegram, G, and A. Bardossy (2013), Downscaling Regional Circulation Model rainfall to gauge sites using recorrelation and circulation pattern dependent quantile–quantile transforms for quantifying climate change, *Journal of Hydrology* 504, 142–159, doi:10.1016/j.jhydrol.2013.09.014.

For instance, if satellite datasets were compared with gridded-based dataset (just for checking, not for the merging procedure), what are the results? I believe that a different picture can be obtained (but I could be totally wrong). Can the authors add some additional discussions on this aspect?

We would not expect a markedly different picture if we had used a gridded gauge-based dataset (e.g., CPC Unified) instead of the GHCN-D station data. This is because in grid cells with gauges, the gridded datasets would generally follow nearly exactly the temporal dynamics of the station data.

2) As hydrologist, I was very interested from the analysis for evaluating MSWEP dataset through hydrological modelling. Particularly, Figure 9a highlights that the overall performance of satellite-only MSWEP dataset is significantly lower than the reanalysis-only and gauge-only MSWEP dataset. Do the authors have an explanation for these large differences?

Thank you. We have added the following statement to the revised m/s: "The overall low performance of the satellite-only MSWEP reflects the lack of tropical catchments in our catchment set."

In Figure 8b, satellite-only performance are better than the reanalysis in the tropical region, but (nearly) always lower than the gauge-based product. As the paper represents also one of the first studies performing a comprehensive assessment, on a global scale, of the three sources of precipitation (satellite, reanalysis, raingauge), it would be interesting to extend the discussion of the obtained results through hydrological modelling. I believe it would be highly of interest to the HESS readership.

We thank the reviewer for his encouraging comment and agree this is the first comprehensive assessment of its kind. We mention in the m/s that the good performance of the gauge-only MSWEP is somewhat misleading and happens because the large majority of the catchments are located in regions with dense $P$ measurement networks (page 17 lines 5–7). However, to further address this comment we added the following text to the revised m/s: "These results once again confirm the complementary nature of satellite and reanalysis data clearly demonstrated in the previous evaluation of $P$ datasets using gauge observations (see Section 4.2)."

---

## Referee Report (RR1)

R. Wartenburger

December 1, 2016

This well-written manuscript describes MSWEP, a multi-source 3-hourly precipitation product. For the first time, the authors successfully attempt to combine satellite, reanalysis and gauge-based precipitation products by performing a 3-step weighted merging procedure. A validation exercise states that MSWEP generally outperforms other data products. Although the manuscript has undergone some substantial improvements since initial submission and I clearly recommend it for final publication in HESS, a few points still need to be addressed.

**Major Comments**

1. Definition of $T_a$ threshold to distinguish between rain and snow (pp. 4, 10, 15): The choice of the temperature threshold is inconsistent: to derive the snowfall fraction map for the CR-based bias correction, the authors use 1°C, but for the merging they use 3°C. It is unclear to me a) what the reason for this difference is and b) how / why these numbers have been chosen (e.g., common practice?). Did the authors attempt to use other thresholds? It would be good to state this at some point in the manuscript.

2. Bias correction based on Q observations (p. 7): The Zhang et al. 2001 correction is only applied for regions with snowfall and/or complex topography. One might argue that this choice could lead to inconsistencies at the transition zones between regions for which the adjustment has been applied versus those where it hasn't been applied. Have the authors tested a set-up where the corrections are always applied? This would allow undercatch corrections in non-mountainous, snow-free

regions (e.g., for catchments located in the mid latitude westerlies, coastal stations affected by stronger wind speeds). In my opinion MSWEP might even gain some quality when applying the corrections to all gauge observations, even though this won't change the persistent P undercatch in the polar regions that the authors have pointed out on page 22. Some readers might also wonder how this would affect the agreement with Adams et al. (2006) shown in Figure 2c.

3. Generation of weight maps (p. 10): The median weight of the 10 most nearby gauges is used to interpolate station weights to grid weights. This can lead to some temporal inhomogeneities at time steps where one or several stations have gaps. This could be at least partly be resolved by some sort of gap-filling prior to computing the weights (e.g., following Andersson et al. 2012, see below). I realize that this could cost some effort, so it might also be a feature to be implemented in a future version of MSWEP. However, the reader should at least know that this issue exists. Andersson, J. C.M., Zehnder, A. J.B., Wehrli, B. and Yang, H. (2012), Improved SWAT Model Performance With Time-Dynamic Voronoi Tessellation of Climatic Input Data in Southern Africa1. JAWRA Journal of the American Water Resources Association, 48: 480–493. doi:10.1111/j.1752-1688.2011.00627.x

4. Generation of weight maps (p. 10): besides the point above, I am also a bit doubtful on the decision to use the 10 most nearby gauges. This will ultimately yield lower median weights as compared to choosing the 10 gauges with the highest correlation. E.g., the very localized precipitation variability of a mountain station near the centre of the 0.25°grid cell might not be well reflected by the evaluated gridded data product, while all other nearby stations (within some cone of influence) have a higher correlation. One could now argue that the correlation value of the station with the local P signal should thus not have an influence on the weight assigned to its associated 0.25°grid cell. Similar issues might arise in the third merging stage (equation in line 16 on page 10) when setting $D_0$ to a constant value of $25\,km$, which in fact depends much on terrain roughness. Of course one should not overcomplicate things, but this should at least be kept in mind.

5. P undercatch in FLUXNET (p. 15): I wonder if it would not be even more useful to evaluate MSWEP against undercatch-corrected FLUXNET data. However, as I am not an expert in this field, I am not aware if this is feasible without loosing the independence criterion and without introducing any substantial additional biases. Undercatch-corrected FLUXNET data would, however, have the potential to state whether the undercatch adjustments in MSWEP are in fact reasonable or not, and it should underline MSWEP's superiority relative to the other data products (also in terms of the absolute bias). If it is not an option to correct the FLUXNET data, the reader would certainly appreciate if the authors could at least elaborate a bit more on this point.

6. Median NSE scores increase with increasing distance to the closest P gauge (p. 17): The authors state (indirectly) that this is primarily due to the dominance of (semi-)arid catchments (which tend to exhibit higher NSE scores) in densely gauged groups. Have the authors applied a test-wise removal of the (semi-)arid catchments or how do they come to this conclusion?

**Minor Comments**

- Dataset choice for MSWEP (p. 8): While it is clear to me how the set of satellite and reanalysis P datasets for use in MSWEP was defined, I miss information on how the gauge-only gridded products (CPC Unified and GPCC) were chosen. I assume the reasoning follows the list of issues pointed out on p. 3, but it is nowhere clearly stated that CPC Unified and GPCC are the only gauge-only datasets that meet all criteria.

- Calculation of $B$ (p. 11): How is the long-term mean defined here? Does it correspond to all time steps available in the observational record? I would appreciate as a reader if this was shortly mentioned in the text.

- Central Asian mountain ranges (p. 12): I would not count the mountain range in Iran to central Asia, maybe better rephrase this to something like "mountain ranges in Central Asia and Iran".

- Figure 11 (p. 24): Why is the colour scale inverted here (wrt. e.g. Figure 7)? I think that most readers will, at the first glance, expect that red colours correspond to high NSE scores.